# Order-Level Attention Similarity Across Language Models: A Latent Commonality

**Jinglin Liang**[1], **Jin Zhong**[1], **Shuangping Huang**[1,2]*,
**Yunqing Hu**[3], **Huiyuan Zhang**[3], **Huifang Li**[4], **Lixin Fan**[5], **Hanlin Gu**[5,6]
[1]South China University of Technology, [2]Pazhou Laboratory,
[3]Zhuzhou CRRC Times Electric Co., [4]China Telecom Research Institute,
[5]WeBank, [6]The Hong Kong University of Science and Technology,
`eeljl@mail.scut.edu.cn, eehsp@scut.edu.cn`

## Abstract

In this paper, we explore an important yet previously neglected question: Do context aggregation patterns across Language Models (LMs) share commonalities? While some works have investigated context aggregation or attention weights in LMs, they typically focus on individual models or attention heads, lacking a systematic analysis across multiple LMs to explore their commonalities. In contrast, we focus on the commonalities among LMs, which can deepen our understanding of LMs and even facilitate cross-model knowledge transfer. In this work, we introduce the Order-Level Attention (OLA) derived from the order-wise decomposition of Attention Rollout and reveal that the OLA at the same order across LMs exhibits significant similarities. Furthermore, we discover an implicit mapping between OLA and syntactic knowledge. Based on these two findings, we propose the Transferable OLA Adapter (TOA), a training-free cross-LM adapter transfer method. Specifically, we treat the OLA as a unified syntactic feature representation and train an adapter that takes OLA as input. Due to the similarities in OLA across LMs, the adapter generalizes to unseen LMs without requiring any parameter updates. Extensive experiments demonstrate that TOA's cross-LM generalization effectively enhances the performance of unseen LMs. Code is available at `https://github.com/jinglin-liang/OLAS`.

## 1 Introduction

With the rapid development of large language models (LMs), their exceptional capabilities have profoundly impacted human society [1]. In practical applications, practitioners often fine-tune models to meet the demands of specific tasks [2, 3]. However, due to the lack of efficient methods for knowledge transfer between different LMs, results fine-tuned on one model cannot be directly reused on another, significantly increasing development costs. Research in knowledge distillation [4, 5] and representation learning [6] suggests that when different models share common representational spaces, efficient knowledge transfer becomes possible. This inspires us to ponder a question: Do pretrained LMs possess commonality that could enable cross-model knowledge transfer?

We approach this through the lens of attention mechanisms. Although LMs differ in architecture, training data, and other factors, mainstream transformer-based LMs rely on attention mechanisms [7] to aggregate context for prediction [8, 9]. Given the similarity in training objectives and attention mechanisms, different LMs trained on large corpora may converge to an optimal attention pattern for the same text, resulting in commonalities in their contextual aggregation behavior.

---

*Corresponding Author

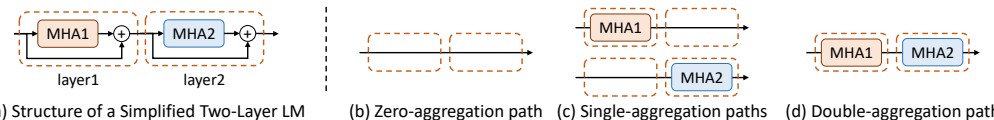

Figure 1: Information flow path decomposition. (a) Simplified LM showing only Multi-Head Attention (MHA) modules (others omitted); (b) Path via residual connections across all layers; (c) Paths through MHA in one layer, residual connections elsewhere; (d) Path through MHA in all layer.

Although some studies [10, 11, 12] have investigated the context aggregation mechanisms within individual LMs, their focus has been on attribution analysis, aiming to quantify the contribution of different tokens to the output [13, 14]. However, focusing solely on attribution analysis of a specific model may lead to an overemphasis on its unique characteristics, without considering the potential commonalities across different LMs. To our knowledge, the commonality of contextual aggregation patterns across different LMs has not yet been explored.

In this work, we unveil this commonality and propose a training-free cross-model knowledge transfer approach based on it. While attention pattern similarities among LMs seem intuitive, inherent differences in layer numbers and attention heads endow their attention weights with distinct meanings, making it challenging to identify attention similarities at the layer or head level. To unify attention weights across models into comparable representations, we propose Order-Level Attention (OLA). Specifically, as shown in Figure 1, we decompose information flow into multiple paths, with the context aggregation effects from paths sharing the same aggregation count being represented as an OLA of that specific order. For instance, first-order OLA captures aggregation effects from paths containing one aggregation step, as shown in Figure 1(c). Mathematically, OLA equates to order-wise decomposition of Attention Rollout [10] (detailed motivations and derivations in §3.1). OLA unifies the meanings of attention across different models, establishing foundations for analyzing attention similarities. Extensive experiments on 12 LMs demonstrate significant similarities in the same-order OLA across different LMs. We refer to this phenomenon as Order-Level Attention Similarity (OLAS). Furthermore, to investigate the linguistic implications of OLA, we conduct experiments demonstrating that syntactic dependencies [15] can be predicted solely from OLA representations using an auxiliary model. This finding suggests that OLA inherently encodes syntactic knowledge of the input text.

Based on these findings, we propose the Transferable OLA Adapter (TOA), enabling cross-LM adapter transfer without requiring tuning. Training-free cross-LM adapter transfer is a valuable task with many potential applications (§4), but poses significant challenges. Adapters process features specific to individual LMs, yet different LMs exhibit divergent feature spaces. Even when training the same model twice with different initialization parameters, the resulting feature spaces remain distinct [6], causing adapters to tightly couple with source LMs and limiting transferability. To address this, we leverage OLA as a unified syntactic representation across LMs and train adapters using OLA for downstream tasks. Since OLA exhibits similarities across LMs, the trained adapter can be directly transferred to other LMs without any parameter updates or training data. We evaluate TOA on four tasks: relation extraction (RE), named entity recognition (NER), dependency parsing (DP), and part-of-speech tagging (POS). Extensive experiments demonstrate significant performance improvements when transferring TOA from a source LM to an unseen target LM. For example, transferring TOA trained on LLaMA3-3B to Qwen2-1.5B elevates Qwen2's relation prediction accuracy from 7.69% (zero-shot baseline) to 34.90%.

In summary, our contributions are as follows:

1) We propose OLA (§3.1), which unifies the attention mechanisms of different LMs into comparable representations.

2) Based on extensive qualitative and quantitative experimental analysis, we propose two key findings: significant similarities in OLA across different LMs (§3.2 and §3.3), and OLA's inherent encoding of syntactic knowledge (§3.4).

3) Building on the above findings, we introduce TOA (§4.1), which enables training-free cross-LM adapter transfer. Extensive experiments demonstrate that transferring TOA trained on a source LM to an unseen target LM significantly enhances its performance (§4.2).

## 2 Related Work

Although some studies have explored the context aggregation mechanisms of LMs and adapter transfer for LMs, we are the first to investigate the commonalities in context aggregation across different LMs and leverage these to achieve cross-LM adapter transfer without requiring tuning.

**Context Aggregation Mechanisms in LMs**. While deep learning has advanced various fields in recent years [16, 17], our understanding of deep models remains limited [18, 19]. This has motivated extensive research on model interpretation, including work on attribution [20, 21] and feature interactions [22, 23]. Among these, some methods focus specifically on analyzing the attention patterns of transformer-based models. For instance, early works [24, 25, 26, 27] aimed to understand the nature of attention by performing intuitive visualizations or statistical analyses in classic models such as Bert [28] and GPT2 [29]. Subsequently, some studies [13, 14, 30, 31] explored the explainability of attention by perturbing attention weights and observing changes in outputs. Additionally, some studies [32, 33] have investigated the identifiability of attention weights. Unlike the aforementioned works that analyze a specific layer or an individual attention head, the study [10] conducts a comprehensive analysis of multi-layer attention and residual connections using matrix multiplication and maximum flow algorithms [34]. Recently, a series of studies based on norm-based methods [35] investigate the contribution of each input token to the output. Specifically, they decompose the model's output into multiple terms, each associated with a particular token, and then estimate the contribution of the token based on the norm of each term or its deviation from the output. Some studies [36, 11] primarily consider the attention blocks responsible for context aggregation, while others [37, 38, 12] further incorporate the feed-forward blocks that map the features of each token into consideration. While existing studies focus on attention explainability and its use in model prediction attribution, the commonalities underlying LM context aggregation mechanisms remain unexplored.

**Adapter Transfer Across LMs**. Freezing the LM's parameters and only training adapters is a common paradigm for applying LMs to specific tasks [39, 3]. These adapters come in various forms, such as LoRA [2] and soft prompts [40]. To reduce the resource consumption associated with repetitive learning, some studies explore adapters transfer. Some works [41, 42, 43] investigate the cross-lingual transfer of adapters, meaning they use adapters trained on a source language in a target language. The work [44] proposes training a delta LM that assembles outputs with the pretrained LM to enable cross-model knowledge transfer. However, their method transfers text-input delta LMs, differing fundamentally from our feature-space adapter transfer approach. The work [45] is the only study that studies cross-model adapter transfer, applying the representation learning [46, 47] method proposed in [6] to soft prompts. However, this approach requires training during transfer and suffers significant performance degradation. To our knowledge, we are the first to achieve training-free cross-model adapter transfer, which allows adapters trained on the source model to be directly applied to the target model without any additional training.

## 3 Order-level Attention

In this section, we first describe the derivation of OLA (§3.1). Then, we present qualitative (§3.2) and quantitative (§3.3) experimental evidence to reveal the phenomenon of OLAS. Finally, we demonstrate the implicit mapping between OLA and syntactic knowledge (§3.4).

### 3.1 Order-Level Decomposition of Attention Rollout

Although different LMs exhibit certain structural variations, typical models such as Bert [28] and Llama [48] feature layers composed of an attention block followed by a feed-forward block, as illustrated in Figure 2. Since the feed-forward block and the normalization layers only perform transformations on the features of individual tokens without aggregating information from other tokens, it is actually the multi-head attention module that facilitates contextual information aggregation at each layer. Additionally, the attention module is paired with a residual connection, creating a shortcut for the information to bypass the attention module. As a result, the context aggregation matrix for the $i$-th layer can be expressed as $(A^{(i)} + I)$, where $A^{(i)} \in \mathbb{R}^{L \times L}$ is the average attention matrix across all heads in the $i$-th layer, $L$ is the token sequence length, and $I$ is the identity matrix representing the shortcut created by the residual connection. By multiplying the matrices of each layer, we obtain the

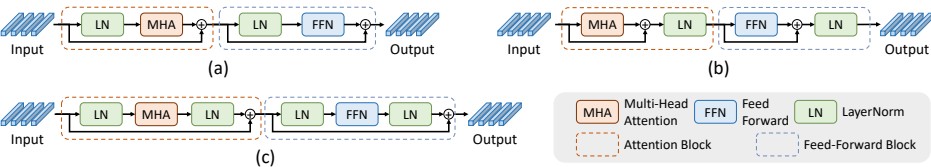

Figure 2: Structure of each layer in typical LMs. (a) Llama and Qwen. (b) Bert, Roberta, and Electra. (c) Gemma.

attention rollout [10], expressed as:

$$\hat{A} = \prod_{i=1}^{N}(A^{(i)} + I),\tag{1}$$

where $\hat{A}$ is the attention rollout, with $N$ as the number of layers in the language model.

However, when we input different texts into LMs and visualize their attention rollout, as shown in Figure 3(a), we observe that the attention rollout exhibits a consistent pattern across different texts, with nearly all attention concentrated on a few less important tokens. More visualizations are presented in §A, showing this phenomenon is prevalent across various LMs. This phenomenon has been observed before and termed "Attention Sinks" [49]. The reason is that the softmax function prevents attention scores from being exactly zero, which forces each token to aggregate information from other tokens. However, when a token has already aggregated sufficient contextual information and no longer needs to aggregate information from other tokens, it offloads its attention to other less important tokens, causing Attention Sinks [49]. This may imply that although LMs consist of many layers, the number of effective aggregation steps may be fewer than the number of layers.

Due to Attention Sinks, the Attention Rollout exhibits similar responses across different texts, lacking distinctiveness. We posit that Attention Sinks occur because each layer's attention module in an LM is paralleled with a residual connection, allowing information to flow both through the attention module and the residual connection. Consequently, as shown in Figure 1, an $N$-layer LM creates $2^N$ potential information pathways. The Attention Rollout represents the contextual aggregation matrix resulting from all these paths. As previously mentioned, the number of effective aggregations is fewer than the number of layers, causing some path components to become ineffective due to excessive aggregation. This results in similar biases and diminishes the distinctiveness of the Attention Rollout. To address this, we separately analyze the contextual aggregation effects from paths with varying aggregation counts, illustrated in Figure 1(b)(c)(d). Mathematically, this involves performing an order-level decomposition of the Attention Rollout, expressed as:

$$\hat{A} = I + \sum_{i=1}^{N} A^{(i)} + \sum_{1 \le i < j \le N} A^{(j)} A^{(i)} + \cdots + A^{(N)} A^{(N-1)} \cdots A^{(1)},\tag{2}$$

where each term corresponds to the sums of contextual aggregation effects across paths with identical aggregation steps. For instance, the 0th-order term is the identity matrix $I$, which means that information flows through residual connections at all layer, as illustrated in Figure 1(b). The first-order term sums effects across all paths containing one aggregation step (i.e., one attention module traversal and $N$-1 residual connections), with $\binom{N}{1}$ such paths, as shown in Figure 1(c). Similarly, the $k$-th order term aggregates effects across $\binom{N}{k}$ paths where aggregation occurs $k$ times.

We normalize each term to obtain the OLA of each order. For example, the first-order OLA is defined as $\hat{A}^{(1)} = \frac{1}{N} \sum_{i=1}^{N} A^{(i)}$, and the Attention Rollout $\hat{A}$ is reformulated as a weighted sum of the OLA:

$$\hat{A} = \sum_{i=0}^{N} \binom{N}{i} \cdot \hat{A}^{(i)},\tag{3}$$

where $\hat{A}^{(i)}$ denotes the $i$-th order OLA.

In summary, given that different LMs share similar optimization objectives and context aggregation mechanisms, it is intuitive that their attention mechanisms exhibit commonalities. To verify this, we

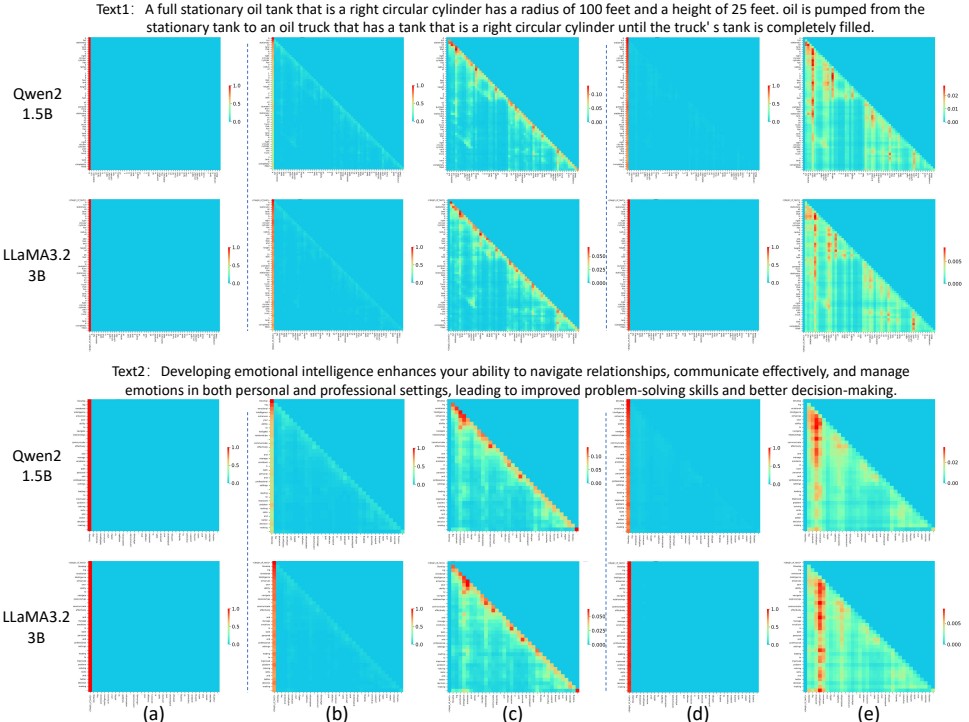

Figure 3: Visualization results of the Attention Rollout and first- and second-order OLA obtained by inputting two texts into Qwen2-1.5b and Llama3.2-3b. (a) Attention Rollout. (b) First-order OLA. (c) First-order OLA with row-wise maximum values set to zero. (d) Second-order OLA. (e) Second-order OLA with row-wise maximum values set to zero.

propose OLA, which unifies attention across LMs into comparable representations with equivalent semantics to enable cross-model comparisons. Below, we analyze OLA's cross-model similarity through quantitative and qualitative perspectives, and explore its linguistic implications.

## 3.2 Qualitative Empirical Evidence of OLAS

To qualitatively analyze OLA similarity across LMs, we input two distinct texts to Qwen2-1.5b and Llama3.2-3b, visualizing Attention Rollout and first-/second-order OLA in Figure 3 (a), (b), and (d). Since extreme maximum values obscure distribution patterns, we mask these values (zeroing maxima for first-/second-order OLA in Figure 3 (c) and (e), and for Attention Rollout in §A), revealing clearer structural features. From the figures, we can derive the following insights: 1) **The same-order OLA from different LMs for the same text is highly similar.** This can be seen by comparing the first and second, as well as the third and fourth rows in Figures 3 (c) and (e). It is also evident in the third-order OLA visualizations in §B. We term this phenomenon Order-Level Attention Similarity (OLAS). 2) **OLA from different texts show a clear distinction.** This can be concluded by comparing the first and third rows, as well as the second and fourth rows in Figures 3 (c) and (e). This suggests that OLA could potentially serve as a feature representation for sentences. 3) **Attention sinks in low-order OLA are less pronounced than in higher-order OLA.** Attention Rollout (weighted sum of all-order OLAs) exhibits the most significant attention sinks, with each row's attention focused on unimportant tokens, such as the Llama's '<bos>' token. First-order OLA exhibits the least sinking, while second-order OLA shows more. This suggests that higher-order OLA may represent ineffective components with similar biases. Comprehensive visualizations for more LMs are in §A and §B.

## 3.3 Quantitative Empirical Evidence of OLAS

To comprehensively quantify the similarity of OLA across different LMs, we design two innovative evaluation methods: the first relies on a visual model to replace human assessment of OLA visual similarity (§3.3.1), the second adopts an image retrieval framework (§3.3.2).

Table 1: Results of quantitative evaluation on cross-model similarity for OLA and baselines based on visual classification model. Entries represent accuracy (unit: %), averaged over three experiments, reflecting source-target LM similarity (higher = more similar). The terms 1st, 2nd, 3rd denote first-, second-, and third-order OLA. Best performance is **bolded**.

(a) CLM Results. Q-1b5, Q-7b, G-2b, G-9b, L-3b, and L-8b denote Qwen2-1.5b, Qwen2-7b, Gemma2-2b, Gemma2-9b, Llama3.2-3b, and Llama3.1-8b.

| Source | L-3b, L-8b, G-2b, G-9b | | L-3b, L-8b, Q-1b5, Q-7b | | Q-1b5, Q-7b, G-2b, G-9b | |
|---|---|---|---|---|---|---|
| Target | Q-1b5 | Q-7b | G-2b | G-9b | L-3b | L-8b |
| Rollout [10] | 27.90 | 7.70 | 52.60 | 26.00 | 66.10 | 59.70 |
| IRNL [36] | 18.50 | 11.90 | 58.10 | 67.20 | 77.20 | 73.60 |
| ALTI [11] | 22.60 | 15.50 | 69.30 | 71.80 | 85.60 | 79.80 |
| 1st | 52.60 | 49.20 | **93.10** | **92.40** | **94.60** | **94.10** |
| 2nd | 67.10 | **49.90** | 89.30 | 86.20 | 90.70 | 91.90 |
| 3rd | **76.50** | 48.90 | 79.90 | 78.80 | 88.60 | 86.70 |

(b) MLM Results. B-b, B-l, R-b, R-l, E-b, and E-l denote Bert-base, Bert-large, Roberta-base, Roberta-large, Electra-base, and Electra-large, respectively.

| Source | R-b, R-l, E-b, E-l | | B-b, B-l, E-b, E-l | | B-b, B-l, R-b, R-l | |
|---|---|---|---|---|---|---|
| Target | B-b | B-l | R-b | R-l | E-b | E-l |
| Rollout | 44.30 | 42.80 | 13.20 | 36.60 | 13.60 | 9.10 |
| IRNL | 60.00 | 4.10 | 20.80 | 15.00 | 55.20 | 38.10 |
| ALTI | 90.30 | 80.30 | 73.80 | 48.10 | 86.80 | 90.30 |
| 1st | **91.90** | **92.40** | **80.40** | **69.90** | **95.20** | **95.80** |
| 2nd | 88.80 | 62.70 | 60.70 | 28.00 | 76.30 | 82.90 |
| 3rd | 80.40 | 67.70 | 38.10 | 31.90 | 58.10 | 68.30 |

### 3.3.1 Quantitative Analysis Based on Visual Model

We employed an image classification model as a proxy for human evaluation to objectively assess the similarity of OLA maps generated by different LMs given identical text inputs. Specifically, we trained an image classifier (ResNet-18 [50]) using OLA maps generated by source LMs, where all OLA maps produced by different LMs for the same text were assigned to the same category, with the optimization objective defined as:

$$\theta^* = \arg\min_{\theta} \mathbb{E}_{(a,i)\sim\mathcal{D}_{train}}\big[\mathcal{L}_{\text{CE}}(F_\theta(a),i)\big], \quad \mathcal{D}_{train} = \big\{(a_i^{(j)},i) \,\big|\, i \in [1..M], j \in [1..S]\big\} \quad (4)$$

where $M$ denotes the count of texts, $S$ the number of source LMs, $a_i^{(j)}$ the OLA map from the $j$-th source LM for the $i$-th text, $\mathcal{D}_{train}$ the training dataset, $\theta$ the classifier parameters, $\mathcal{L}_{\text{CE}}$ the cross-entropy loss, and $F_\theta(a)$ the classifier's predicted text index for OLA map $a$. The trained classifier was evaluated on the dataset $\mathcal{D}_{test}$ composed of OLA generated by target LMs on the same set of texts, where $\mathcal{D}_{test} = \big\{(\tilde{a}_i,i) \,\big|\, i \in [1..M]\big\}$, with $\tilde{a}_i$ denoting the OLA produced by the target LM for the i-th text. Higher accuracy indicates stronger alignment between source and target LMs' OLA, as the classifier more reliably classifies their OLA generated for the same text into the same category. Experiments were conducted on 12 LMs, including 6 Causal Language Models (CLMs) and 6 Masked Language Models (MLMs), detailed descriptions of these LMs are provided in §C. Further implementation details, including dataset construction and preprocessing, are provided in §F.

**Baselines.** To analyze whether the performance of existing context aggregation analysis methods exhibits similarities across different LMs, we also validated them within the experimental framework. The methods include Attention Rollout [10], IRNL [36], and ALTI [11]. Among these, Attention Rollout is the most similar to our method, while IRNL and ALTI are common norm-based attribution methods focused on attention block analysis. Additionally, IRNL and ALTI require deriving the expression between the model's output and input tokens. However, their derivations were only conducted for MLMs and cannot be directly applied to CLMs due to structural differences. Therefore, we derive the expressions for CLMs and present the process in §D.

**Main Results.** From the Table 1, we can draw the following insights: 1) As concluded in §3.2, **the OLA obtained from the same text across different LMs exhibits significant similarity, while OLA from different texts shows clear distinctions.** In experiments with both MLMs and CLMs, the first-, second-, and third-order OLA achieved high classification accuracy, particularly the first-order OLA, which exceeded 90% accuracy under multiple settings. The fact that OLA can be used for classification indicates that OLA from different texts are distinguishable. Furthermore, the ability of a classifier trained on the source LMs' OLA to generalize to the target LM suggests that OLA across the source and target LMs are highly similar. 2) **Existing methods also exhibit similar performance across different LMs, but not as prominently as the simpler OLA.** This may arise because Attention Rollout incorporates higher-order OLA components with weaker distinguishability. Other norm-based methods for attribution analysis primarily focus on the relationships between individual

Table 2: Retrieval-based quantitative evaluation of first-order OLA cross-model similarity. Row headers denote source LMs. Column headers denote target LMs. Entries indicate evaluation metrics Hits@1 / Hits@5 (unit: %).

(a) CLM Results.

| Src\Tgt | Q-1b5 | G-2b | L-3b |
|---|---|---|---|
| Q-1b5 | - | 83.60 / 89.40 | 95.90 / 97.00 |
| G-2b | 83.20 / 89.30 | - | 95.30 / 97.10 |
| L-3b | 92.90 / 96.10 | 94.10 / 96.50 | - |

(b) MLM Results.

| Src\Tgt | B-b | R-b | E-b |
|---|---|---|---|
| B-b | - | 51.90 / 58.80 | 91.60 / 94.90 |
| R-b | 75.90 / 83.90 | - | 71.70 / 80.20 |
| E-b | 92.40 / 96.00 | 67.40 / 72.90 | - |

Table 3: Results of syntactic dependency parsing using OLA predicted by LMs. Entries indicate UAS/LAS (unit: %). Best performance is **bolded**.

(a) CLM Results.

| LMs | Q-1b5 | G-2b | L-3b |
|---|---|---|---|
| Rollout | 50.53/29.79 | 44.24/22.04 | 53.77/35.57 |
| 1st | **63.58/48.24** | **62.25/45.95** | **62.98/48.19** |
| 2st | 60.58/43.90 | 57.28/38.88 | 58.93/42.94 |
| 3rd | 55.19/36.82 | 51.89/32.25 | 51.00/33.35 |

(b) MLM Results.

| LMs | B-b | R-b | E-b |
|---|---|---|---|
| Rollout | 46.20/30.69 | 35.77/17.94 | 50.35/34.02 |
| 1st | **81.29/72.16** | **80.00/70.44** | **81.23/72.63** |
| 2st | 72.86/61.05 | 72.68/60.10 | 77.47/66.78 |
| 3rd | 66.44/53.17 | 36.99/18.67 | 50.72/33.90 |

tokens in the feature space, which, compared to the contextual aggregation patterns we emphasize, are more reflective of the model's individual characteristics. 3) **The similarity of OLA varies across different source-target LMs combinations.** This suggests that the similarity between different LMs is relative and influenced by various factors during training, such as data and architecture. OLA has the potential to serve as an indicator for evaluating the similarity between LMs.

**Controlled Experiments.** To verify the reproducibility of our findings across diverse data settings, we conducted controlled experiments on dataset and preprocessing. The OLAS phenomenon persists under varying data configurations, demonstrating its robustness (results and analysis in §H). To ensure our observations reflect inherent properties of LMs (i.e., the commonality of contextual aggregation patterns learned from large corpora), we perturbed the model parameters and observed that the OLAS phenomenon disappeared in the perturbed models. This indicates OLAS is intrinsically tied to pre-trained model parameters, not experimental biases or data artifacts (results and analysis in §I).

### 3.3.2 Quantitative Analysis Based on Image Similarity Retrieval

We propose an image retrieval-based quantitative evaluation method for OLA similarity. Specifically, we feed the text from Section 3.3.1 into the source LM and target LM to generate corresponding OLA maps. Using the target LM's OLA maps as queries, we compute SSIM [51] (a standard image similarity metric) between each query and all source LM's OLA, then rank the results by SSIM scores. Retrieval correctness is determined by whether the ground-truth candidate appears in the retrieved results. The ground-truth is defined as the source LM's OLA whose original text matches the query OLA's original text. We evaluate Hits@1 (the probability of the correct source LM OLA ranking first) and Hits@5 (the probability of the correct OLA appearing in the top five), with first-order results shown in Table 2 and second- and third-order results shown in the §J (Table 12). From the tables, we observe remarkably high retrieval success rates. For example, in the first-order CLM results, even the lowest Hits@5 surpasses 89%, while the highest exceeds 97%. Though MLM performance is weaker than CLM, it remains substantial. These findings further support the OLAS phenomenon.

### 3.4 Relation between OLA and Syntactic Knowledge

The OLAS phenomenon suggests a unified attention pattern across different LMs in OLA. To explore the nature of OLA, we designed an experiment to investigate whether OLA contains syntactic knowledge. Specifically, we utilize the OLA of training texts to train an additional syntactic dependency parsing network, where the input is the OLA and the target is the corresponding syntactic dependency annotations. After training, we evaluate the accuracy of this network on the OLA of test texts. If

the network can successfully predict the original text's syntactic dependencies using only OLA, it suggests that OLA encodes syntactic knowledge. More implementation details can be found in §G.

We conducted experiments on first-, second-, and third-order OLA, respectively. Additionally, since Attention Rollout is equivalent to the weighted sum of OLA across all orders, we included it in our analysis to investigate whether higher-order OLA encode syntactic knowledge. From the results presented in Table 3, we derive the following insights: 1) **OLA encodes syntactic knowledge.** Using first-, second-, and third-order OLA to predict syntactic dependencies achieves promising performance. Notably, first-order OLA achieves over 80% Unlabeled Attachment Score (UAS) across all MLMs and over 60% UAS for CLMs. 2) **Lower-order OLA exhibit more prominent syntactic features than higher-order OLA.** Across all LMs, we observe a consistent trend where higher-order OLA yield lower performance. Furthermore, Attention Rollout (as the aggregation of all-order OLA) exhibits significantly lower accuracy compared to lower-order OLA, this suggests that its higher-order components contain less prominent syntactic features.

## 4 Training-free Cross-LM Adapter Transfer

### 4.1 Transferable OLA Adapter

Due to the large number of parameters in LMs, directly fine-tuning them is often prohibitively expensive. As a result, freezing the parameters of the LMs and training an adapter tailored to downstream tasks has become a common approach for applying LMs to specific tasks [39, 3]. However, adapters are typically tied to the specific LMs they are trained on, which limits their flexibility and reusability. To address this limitation, we investigate how to transfer adapters across LMs. This is a valuable question with many potential applications. For example, we could first train an adapter on a smaller model and then transfer it to a larger model, significantly reducing the resource cost of tuning the adapter for the larger model. In another example, we could train an adapter on an open-source model and transfer it to a closed-source model. This enables the closed-source model to learn knowledge from the data while keeping the model and data isolated, thereby protecting the privacy of both the model and the data.

However, this is a challenging problem, especially in the context of training-free transfer. This difficulty arises because adapters are designed to process the features of LMs, and different LMs often have significantly different feature distributions and dimensions. As a result, the adapter becomes tightly coupled with the source model, making it difficult to directly transfer it to other models. To the best of our knowledge, no existing work has achieved training-free cross-LM adapter transfer.

Inspired by our findings that OLA from different LMs exhibits similarity and that OLA encodes syntactic knowledge, we propose the Transferable OLA Adapter (TOA), which enables adapter transfer across models without requiring training. Specifically, we treat OLA as a unified syntactic feature representation across models and train an adapter that takes OLA as input for downstream tasks. Due to the cross-model

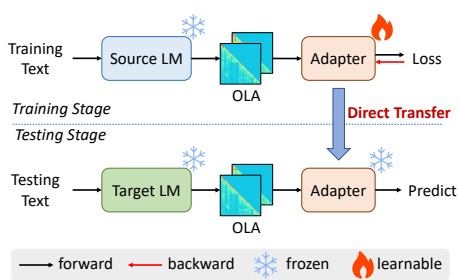

Figure 4: Overview of TOA. In the training phase, the source LM is frozen and an adapter is trained for the downstream task using OLA as input. In the testing phase, the adapter is directly transferred to the target LM.

similarity of OLA, the trained adapter can be directly transferred to other LMs without requiring additional training, as illustrated in Figure 4. In the experiments reported later, we use stacked first-and second-order OLA as input features for the adapter. However, this is not the only option. Other configurations, such as using only first-order OLA or combining different orders, can be chosen based on task requirements.

### 4.2 Experiments

We evaluated the cross-model transfer capability of TOA on four foundational NLP tasks: Relation Extraction (RE) [52], Named Entity Recognition (NER), Dependency Parsing (DP), and Part-of-

Table 4: Cross-Model Transferability of TOA on the RE Task. Column headers indicate source LMs, row headers indicate target LMs, and entries represent relation prediction accuracy (unit: %). Best performance is **bolded**; scores exceeding the zero-shot baseline are underlined.

(a) CLM Results.

| Src\Tgt | Q-1b5 | Q-7b | G-2b | G-9b | L-3b | L-8b |
|---|---|---|---|---|---|---|
| Zero-shot | | | | | | |
| - | 7.69 | 8.58 | 5.01 | 18.22 | 14.65 | 12.99 |
| TOA (Ours) | | | | | | |
| Q-1b5 | **34.90** | 26.33 | 30.95 | 25.98 | 31.08 | 29.46 |
| Q-7b | 27.58 | **31.48** | 25.25 | 21.60 | 25.41 | 25.44 |
| G-2b | 21.17 | 19.92 | **34.73** | 23.33 | 23.49 | 22.64 |
| G-9b | 18.63 | 17.42 | 22.12 | **26.28** | 20.99 | 20.73 |
| L-3b | 30.49 | 22.35 | 33.49 | 22.19 | **35.57** | 33.03 |
| L-8b | 28.24 | 22.28 | 30.03 | 25.80 | 32.03 | **33.43** |

(b) MLM Results.

| Src\Tgt | B-b | B-l | R-b | R-l | E-b | E-l |
|---|---|---|---|---|---|---|
| Zero-shot | | | | | | |
| - | 2.69 | 0.48 | 0.04 | 7.18 | 5.78 | 0.04 |
| TOA (Ours) | | | | | | |
| B-b | **36.19** | 29.90 | 23.90 | 23.60 | 25.70 | 28.28 |
| B-l | 22.13 | **32.29** | 19.33 | 23.16 | 18.63 | 17.45 |
| R-b | 25.63 | 18.70 | **32.63** | 21.61 | 26.40 | 22.50 |
| R-l | 25.59 | 28.98 | 25.15 | **32.73** | 24.12 | 25.15 |
| E-b | 36.01 | 26.99 | 31.96 | 25.77 | **41.27** | 36.97 |
| E-l | 31.85 | 31.89 | 30.63 | 24.15 | 32.99 | **38.18** |

Table 5: Cross-Model Transferability of TOA on the NER Task. Entries represent F1 score (unit: %). Best performance is **bolded**; scores exceeding the zero-shot baseline are underlined.

(a) CLM Results.

| Src\Tgt | Q-1b5 | Q-7b | G-2b | G-9b | L-3b | L-8b |
|---|---|---|---|---|---|---|
| Zero-shot | | | | | | |
| - | 5.35 | 28.21 | 1.45 | 53.82 | 13.24 | 22.12 |
| TOA (Ours) | | | | | | |
| Q-1b5 | **53.81** | 30.99 | 21.08 | 9.58 | 29.24 | 26.94 |
| Q-7b | 36.28 | **54.85** | 15.56 | 10.01 | 26.17 | 28.30 |
| G-2b | 23.12 | 12.47 | **54.53** | 24.31 | 22.64 | 15.80 |
| G-9b | 12.80 | 9.94 | 31.15 | **55.36** | 16.65 | 12.74 |
| L-3b | 27.48 | 21.79 | 24.79 | 14.05 | **54.51** | 45.24 |
| L-8b | 27.93 | 25.33 | 20.63 | 16.34 | 47.84 | **51.68** |

(b) MLM Results.

| Src\Tgt | B-b | B-l | R-b | R-l | E-b | E-l |
|---|---|---|---|---|---|---|
| Zero-shot | | | | | | |
| - | 0.00 | 0.00 | 0.00 | 0.00 | 0.00 | 0.00 |
| TOA (Ours) | | | | | | |
| B-b | **68.47** | 46.00 | 29.74 | 16.62 | 39.14 | 39.74 |
| B-l | 36.40 | **68.74** | 9.69 | 14.14 | 14.54 | 13.21 |
| R-b | 41.66 | 36.02 | **60.99** | 40.60 | 39.85 | 42.90 |
| R-l | 31.43 | 36.12 | 40.31 | **64.60** | 20.50 | 27.35 |
| E-b | 36.28 | 25.53 | 20.83 | 12.21 | **65.14** | 46.87 |
| E-l | 24.62 | 23.98 | 17.71 | 8.38 | 34.51 | **67.08** |

Speech Tagging (POS). Since TOA is directly transferred to the target LM without any training (it is trained solely on the source LM), we compared its performance against the zero-shot capability of the target LM to assess the utility of TOA. Specifically, the TOA transfer process involves training an adapter using the OLA generated by the source LM on the training set and evaluating its performance on the OLA produced by the target LM on the test set. For zero-shot evaluation, we guided LMs' predictions using manually designed prompts, with prompt templates and implementation details provided in §K. The results for RE and NER are presented in Tables 4 and 5, while those for DP and POS are included in §L (Tables 15 and 16). Detailed implementation for TOA transfer (adapter architectures, datasets, metrics) are elaborated in §G.

**Main Results.** From Tables 4, 5, 15, and 16, we derive the following insights: 1) **In most settings, TOA consistently surpasses the zero-shot baseline, demonstrating its practical utility.** For MLMs, TOA surpasses zero-shot performance across all source-target LM pairs on all four tasks. For CLMs, TOA exceeds zero-shot performance in all source-target LM pairs for RE and DP tasks, but underperforms zero-shot in a small fraction of NER and POS cases (less than 6% of total scenarios). These underperforming cases primarily involve larger, high-capacity target LMs (e.g., Gemma2-9B). This suggests that TOA may not improve performance for large, high-capacity models on some tasks but delivers significant gains for smaller models like Qwen2-1.5B and Llama3.2-3B. 2) **Cross-model transfer incurs insignificant performance degradation compared to self-transfer.** The diagonal entries in the tables represent self-transfer results, where the source and target models are identical (i.e., adapters are trained and tested on the same model). These entries serve as an upper-bound baseline for TOA's transfer capability. Remarkably, cross-model performance remains close to this baseline. For example, transferring TOA from BERT-base to BERT-large achieves 29.90% accuracy on the RE task, reaching 93% of BERT-large's self-transfer performance (32.29%). This indirectly

supports the existence of the OLAS phenomenon. 3) **TOA achieves stronger performance with MLMs than CLMs.** This may stem from the bidirectional attention in MLMs, which captures richer contextual information compared to the unidirectional attention in CLMs. 4) **TOA performs better on syntax-dependent tasks (DP, POS) than on semantics-driven tasks (RE, NER).** This suggests that OLA primarily encodes syntactic structures rather than semantic knowledge.

## 5 Conclusion

In this work, we introduced a novel perspective for analyzing LMs by focusing on the commonalities in context aggregation patterns. We revealed the significant similarity in OLA across different LMs, marking a key discovery in understanding the shared mechanisms of LMs. Furthermore, we explored the linguistic implications of OLA and found that it encodes syntactic knowledge. Building on these findings, we proposed the TOA, achieving training-free cross-LM adapter transfer. Extensive experiments demonstrate that TOA trained on other LMs can be transferred to unseen LMs to enhance their performance, yielding promising results.

## 6 Acknowledgement

The research is partially supported by National Natural Science Foundation of China (No. 62576139, 62176093, 61673182), National Key Research and Development Program of China (No.2023YFC3502900), Guangdong Emergency Management Science and Technology Program (No.2025YJKY001).

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

# A Visualization of Attention Rollout

We present the visualization results of Attention Rollout on six LMs, including three MLMs and three CLMs. Specifically, we show the visualizations of Attention Rollout obtained from different text inputs in Figure 5 (a) and (c), and the results after setting outliers in Attention Rollout to zero in Figure 5 (b) and (d). Outliers are defined as values greater than the mean of the row plus three times the standard deviation. The detailed calculation process is provided in §F.

From the figures, we observe that Attention Rollout exhibits a significant attention sink phenomenon, with attention being highly concentrated on a few unimportant tokens. Additionally, the Attention Rollout of different sentences lacks distinguishability, indicating that it is challenging to use it as a feature representation for sentences.

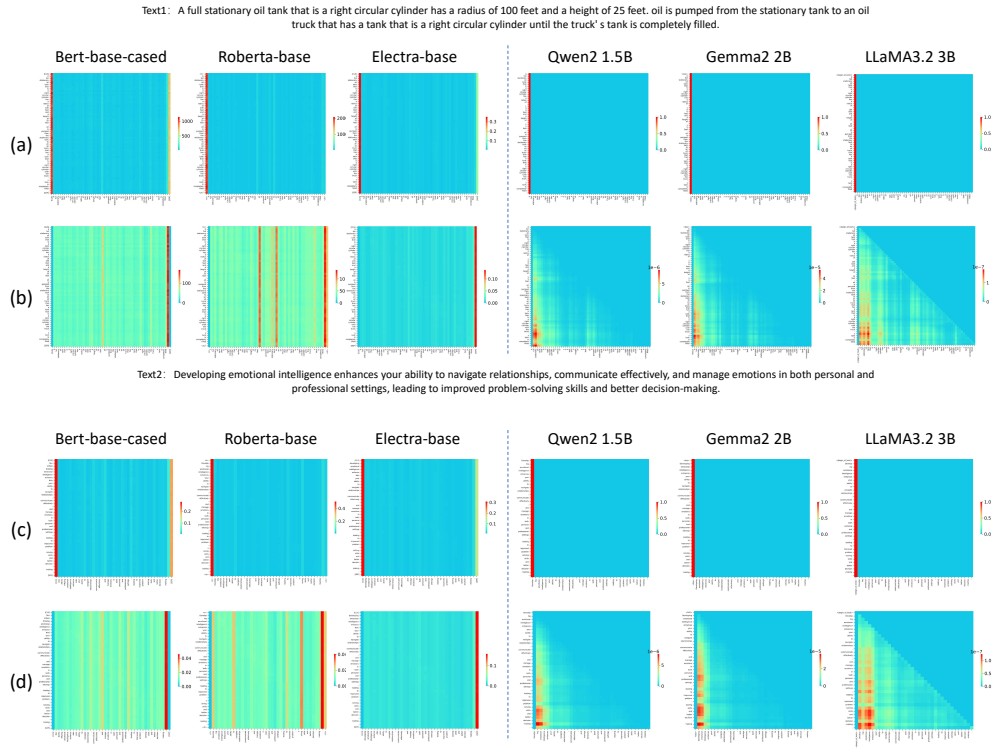

Figure 5: Visualization results of Attention Rollout obtained from two texts input into Bert-base-cased, Roberta-base, Electra-base, Qwen2-1.5b, Gemma2-2b, and Llama3.2-3b. (a) Attention Rollout for Text 1. (b) Attention Rollout for Text 1 with outlier values set to zero. (c) Attention Rollout for Text 2. (d) Attention Rollout for Text 2 with outlier values set to zero.

# B Visualization of OLA

We visualize the first-, second-, and third-order OLA obtained by inputting two texts into LMs, as well as the results after setting outlier values in the OLA to zero. The visualizations for CLMs are presented in Figure 6, and those for MLMs are shown in Figure 7. Outliers are defined as values greater than the mean of the row plus three times the standard deviation. The detailed calculation process is provided in §F. The comprehensive visualization results further support our conclusions in §3.2.

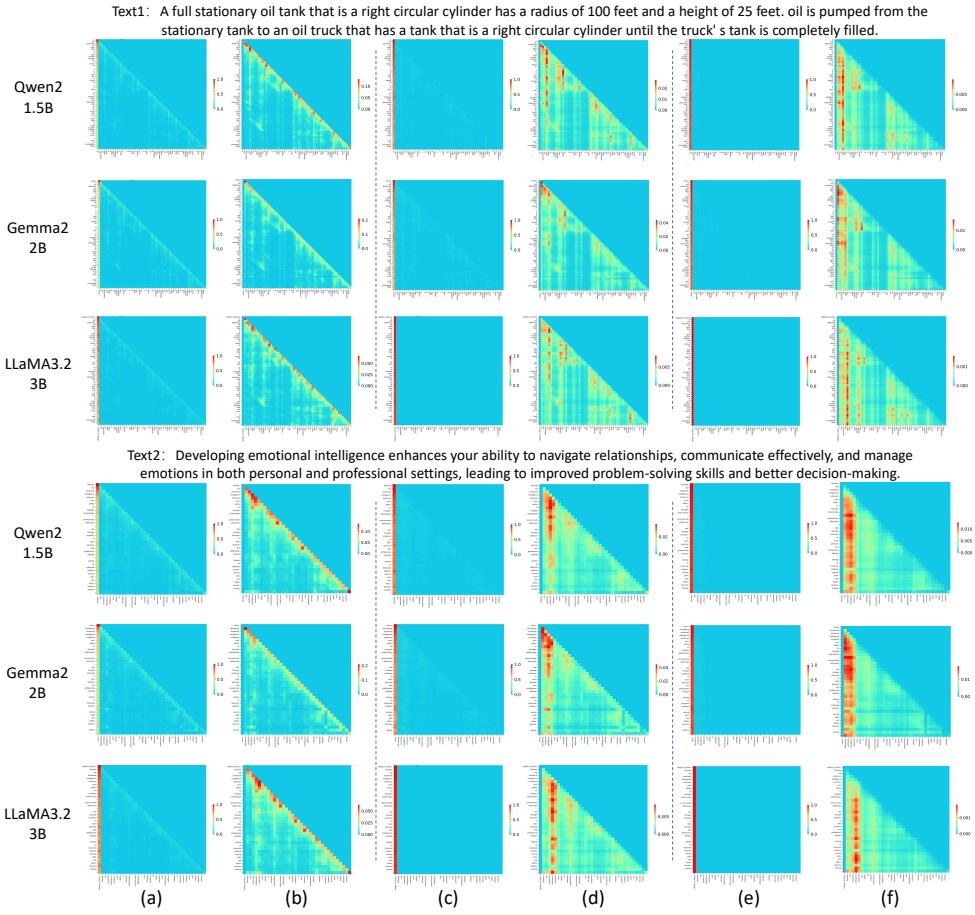

Figure 6: Visualization results of first-order, second-order and third-order OLA obtained by inputting two texts into Qwen2-1.5b, Gemma2-2b and Llama3.2-3b. (a) First-order OLA. (b) First-order OLA with outlier values set to zero. (c) Second-order OLA. (d) Second-order OLA with outlier values set to zero. (e) Third-order OLA. (f) Third-order OLA with outlier values set to zero.

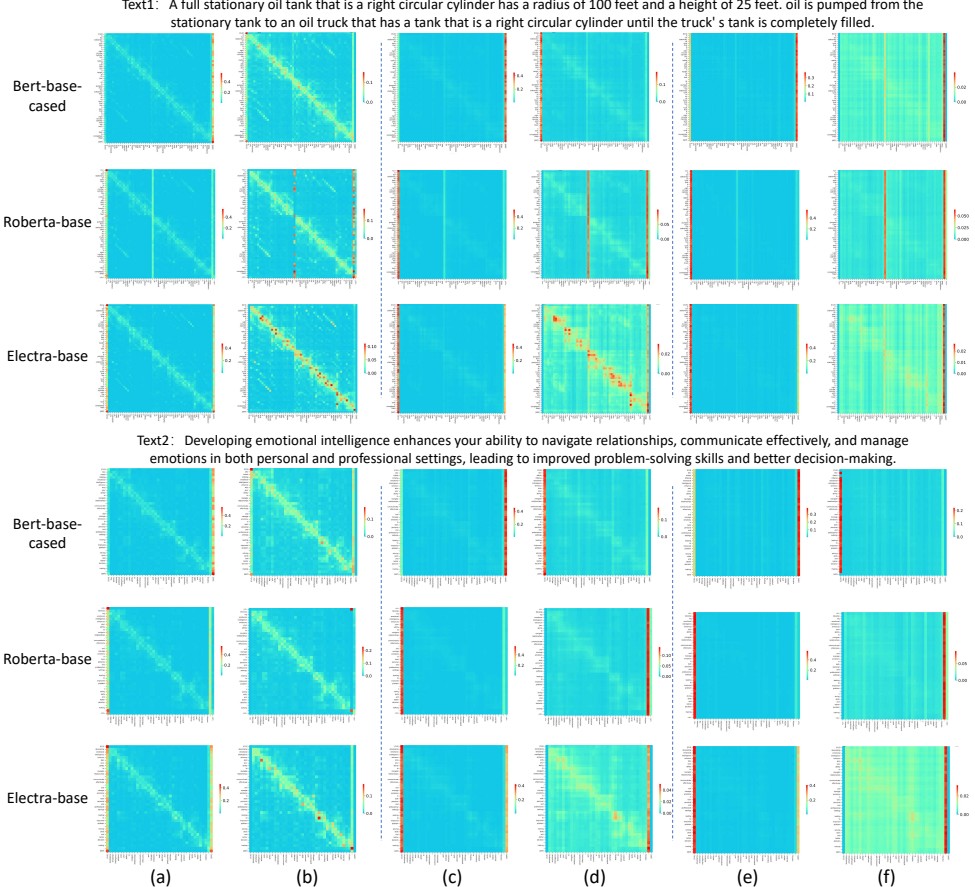

Figure 7: Visualization results of first-order, second-order and third-order OLA obtained by inputting two texts into Bert-base-cased, Roberta-base and Electra-base. (a) First-order OLA. (b) First-order OLA with outlier values set to zero. (c) Second-order OLA. (d) Second-order OLA with outlier values set to zero. (e) Third-order OLA. (f) Third-order OLA with outlier values set to zero.

## C  Language Models we used

We conducted experiments on twelve commonly used LMs, including six CLMs: Qwen2-1.5B[2], Qwen2-7B[3], Gemma2-2B[4], Gemma2-9B[5], Llama3.2-3B[6], and Llama3.1-8B[7], and six MLMs: Bert-base-cased[8], Bert-large-cased[9], Roberta-base[10], Roberta-large[11], Electra-base[12], and Electra-large[13]. For Electra, which consists of a generator and a discriminator, we use the generator because its training objective aligns with traditional MLMs such as Bert and Roberta. The architectural hyperparameters, training data size, and vocabulary size of these models are detailed as Table 6 and Table 7.

Table 6: Details of masked language models.

| Models | Bert-base-cased | Bert-large-cased | Roberta-base | Roberta-large | Electra-base | Electra-large |
|---|---|---|---|---|---|---|
| Hidden Size | 768 | 1,024 | 768 | 1,024 | 768 | 1,024 |
| Layers | 12 | 24 | 12 | 24 | 12 | 24 |
| Attention Heads | 12 | 16 | 12 | 16 | 4 | 4 |
| Head Size | 64 | 64 | 64 | 64 | 192 | 256 |
| Vocabulary Size | 28,996 | 28,996 | 50,265 | 50,265 | 30,522 | 30,522 |
| Trained Tokens | 3.3B | 3.3B | - | - | 3.3B | 33B |

Table 7: Details of causal language models.

| Models | Qwen2-1.5B | Qwen2-7B | Gemma2-2B | Gemma2-9B | Llama3.2-3B | Llama3.1-8B |
|---|---|---|---|---|---|---|
| Hidden Size | 1,536 | 3,584 | 2,304 | 3,584 | 3,072 | 4,096 |
| Layers | 28 | 28 | 26 | 42 | 28 | 32 |
| Query Heads | 12 | 28 | 8 | 16 | 24 | 32 |
| Key Value Heads | 2 | 4 | 4 | 8 | 8 | 8 |
| Head Size | 128 | 128 | 256 | 256 | 128 | 128 |
| Vocabulary Size | 151,936 | 152,064 | 256,000 | 256,000 | 128,256 | 128,256 |
| Trained Tokens | 7T | 7T | 2T | 8T | 9T | 15T |

---

[2] Qwen2-1.5B: `https://huggingface.co/Qwen/Qwen2-1.5B-Instruct`

[3] Qwen2-7B: `https://huggingface.co/Qwen/Qwen2-7B-Instruct`

[4] Gemma2-2B: `https://huggingface.co/google/gemma-2-2b-it`

[5] Gemma2-9B: `https://huggingface.co/google/gemma-2-9b-it`

[6] Llama3.2-3B: `https://huggingface.co/meta-llama/Llama-3.2-3B-Instruct`

[7] Llama3.1-8B: `https://huggingface.co/meta-llama/Llama-3.1-8B-Instruct`

[8] Bert-base-cased: `https://huggingface.co/google-bert/bert-base-cased`

[9] Bert-large-cased: `https://huggingface.co/google-bert/bert-large-cased`

[10] Roberta-base: `https://huggingface.co/FacebookAI/roberta-base`

[11] Roberta-large: `https://huggingface.co/FacebookAI/roberta-large`

[12] Electra-base: `https://huggingface.co/google/electra-base-generator`

[13] Electra-large: `https://huggingface.co/google/electra-large-generator`

# D  Derivation of Norm-based Decomposition for CLMs

The norm-based method requires decomposing the features of the language model into terms associated with each token, and this decomposition process is highly tied to the model's structure. Since ALTI and IRNL only derive the decomposition for MLMs, and due to the structural differences between CLMs and MLMs, the decomposition expression in ALTI and IRNL cannot be directly applied to CLMs. To adapt ALTI and IRNL to CLMs, we derive the decomposition for CLMs.

## D.1  Decomposition for Llama and Qwen

The structures of Llama and Qwen are shown in Figure 2(a). It is important to note that their LN module is not the standard LayerNorm, but instead Root Mean Square Layer Normalization (RMSLN) [53], expressed as:

$$\bar{x}_i = \mathrm{RMSLN}(x_i) = \frac{\gamma}{\mathrm{RMS}(x_i)} x_i, \quad \text{where } \mathrm{RMS}(x_i) = \sqrt{\frac{1}{d} \sum_{j=1}^{d} x_i^{(j)^2}}. \tag{5}$$

Here, $x_i \in \mathbb{R}^{1 \times d}$ represents the input feature of the $i$-th token, $d$ is the hidden dimension of the language model, $\bar{x}_i$ represents the output feature of the RMSLN for the $i$-th token, $\mathrm{RMS}(x_i)$ denotes the root mean square of $x_i$, denotes the $i$-th element of, and $\gamma$ is a learnable parameter.

Then, we analyze the decomposition of MHA, which is expressed as follows:

$$\hat{x}_i = cat(z_i^1, z_i^1, ..., z_i^H) \cdot W_o, \tag{6}$$

$$= \sum_{h}^{H} z_i^h \cdot W_o^h,$$

$$= \sum_{h}^{H} \left( \sum_{j}^{J} A_{ij}^h \cdot \bar{x}_j \cdot \hat{W}_v^h \right) \cdot W_o^h,$$

$$= \sum_{j}^{J} \sum_{h}^{H} A_{ij}^h \cdot \bar{x}_j \cdot \hat{W}_v^h \cdot W_o^h,$$

$$= \sum_{j}^{J} \left( \frac{\gamma}{\mathrm{RMS}(x_j)} \sum_{h}^{H} A_{ij}^h \cdot x_j \cdot \hat{W}_v^h \cdot W_o^h \right).$$

Here, $\hat{x}_i$ represents the feature of the $i$-th token in the output of MHA, $A_{ij}^h$ denotes the $i$-th row and $j$-th column element of the attention matrix for the $h$-th head, $z_i^h$ represents the output after aggregating the values of other tokens for the $i$-th token in the $h$-th head. $W_v \in \mathbb{R}^{d \times (M \times E)}$ is the value projection matrix of MHA, where $M$ is the number of heads for the keys and values, and $E$ is the dimension of each head. $\hat{W}_v \in \mathbb{R}^{d \times (H \times E)}$ represents the matrix obtained by replicating $W_v$ of $H/M$ times and then concatenating, and $H$ is the number of heads for the query. $\hat{W}_v^h \in \mathbb{R}^{d \times E}$ represents the $h$-th block of $W_v$. $W_o \in \mathbb{R}^{(H \times E) \times d}$ represents the output projection matrix of MHA, and $W_o^h \in \mathbb{R}^{E \times d}$ represents the $h$-th block of $W_o$.

Since there is a residual connection in parallel with MHA and LN, the output of the attention block can be expressed as:

$$y_i = \hat{x}_i + x_i = \sum_{j}^{J} \left( \frac{\gamma}{\mathrm{RMS}(x_j)} \sum_{h}^{H} A_{ij}^h \cdot x_j \cdot \hat{W}_v^h \cdot W_o^h \right) + x_i, \tag{7}$$

where $y_i$ represents the output feature of the $i$-th token in the attention block.

**In summary, the decomposition expressions for Llama and Qwen are as follows:**

$$T_i(x_j) = \begin{cases} \frac{\gamma}{\mathrm{RMS}(x_j)} \sum_{h}^{H} A_{ij}^h \cdot x_j \cdot \hat{W}_v^h \cdot W_o^h + x_j & \text{if } i = j \\ \frac{\gamma}{\mathrm{RMS}(x_j)} \sum_{h}^{H} A_{ij}^h \cdot x_j \cdot \hat{W}_v^h \cdot W_o^h & \text{if } i \neq j, \end{cases} \tag{8}$$

where $T_i(x_j)$ represents the term associated with $x_j$ obtained after decomposing the output feature of the attention block for the $i$-th token.

## D.2 Decomposition for Gemma

As shown in Figure 2(c), Gemma2's differences from Llama and Qwen are mainly in two aspects: 1) Its LN is also RMSLN, but its learnable parameters differ, as it includes a fixed bias of size 1; 2) Its attention block has two LNs, one before and one after MHA. Therefore, the expression for the first LN is:

$$\bar{x}_i = \text{RMSLN}(x_i) = \frac{1 + \gamma_1}{\text{RMS}(x_i)} x_i, \quad \text{where RMS}(x_i) = \sqrt{\frac{1}{d} \sum_{j=1}^{d} x_i^{(j)^2}}. \tag{9}$$

Here, $\gamma_1$ represents the learnable parameter of the first LN.

The decomposition of MHA is as follows:

$$\hat{x}_i = \sum_{j}^{J} \left( \frac{1 + \gamma_1}{\text{RMS}(x_j)} \sum_{h}^{H} A_{ij}^h \cdot x_j \cdot \hat{W}_v^h \cdot W_o^h \right). \tag{10}$$

The output of the attention block can be expressed as:

$$y_i = \text{RMSLN}(\hat{x}_i) + x_i = \text{RMSLN} \left( \sum_{j}^{J} \left( \frac{1 + \gamma_1}{\text{RMS}(x_j)} \sum_{h}^{H} A_{ij}^h \cdot x_j \cdot \hat{W}_v^h \cdot W_o^h \right) \right) + x_i \tag{11}$$

$$= \sum_{j}^{J} \left( \frac{(1 + \gamma_1) \cdot (1 + \gamma_2)}{\text{RMS}(\hat{x}_i) \cdot \text{RMS}(x_j)} \sum_{h}^{H} A_{ij}^h \cdot x_j \cdot \hat{W}_v^h \cdot W_o^h \right) + x_i,$$

where $\gamma_1$ represents the learnable parameter of the second LN.

**In summary, the decomposition expressions for Gemma is as follows:**

$$T_i(x_j) = \begin{cases} \frac{(1+\gamma_1) \cdot (1+\gamma_2)}{\text{RMS}(\hat{x}_i) \cdot \text{RMS}(x_j)} \sum_{h}^{H} A_{ij}^h \cdot x_j \cdot \hat{W}_v^h \cdot W_o^h + x_i & \text{if } i = j \\ \frac{(1+\gamma_1) \cdot (1+\gamma_2)}{\text{RMS}(\hat{x}_i) \cdot \text{RMS}(x_j)} \sum_{h}^{H} A_{ij}^h \cdot x_j \cdot \hat{W}_v^h \cdot W_o^h & \text{if } i \neq j. \end{cases} \tag{12}$$

# E  Datasets

This paper employs the following five datasets, which are introduced below.

**CoNLL2012** [54]. It is a multilingual, multi-genre, and multi-task dataset. It supports tasks such as Named Entity Recognition (NER), part-of-speech tagging, coreference resolution, and more. We utilize the corpus of this dataset in the quantitative analysis experiments of OLAS (§3.3.1) and employ both the corpus and its NER annotations in the experiments exploring the cross-model transferability of TOA on the NER task (§4.2). The dataset contains 1,940 documents in the training set and 222 documents in the test set. Each document has an average of 39 sentences.

**UD-English-EWT v2.15** [15]. It is a subset of the Universal Dependencies project[14], serves as a key resource for dependency parsing. This corpus primarily consists of English texts sourced from web-based content, including blogs, reviews, and social media posts. We utilize this dataset in the experiments investigating the relationship between OLA and syntactic knowledge (§3.4), and the experiments exploring the cross-model transferability of TOA on the DP task (§4.2). Its training set contains 12,544 sentences, and the test set contains 2,077 sentences.

**SemEval-2010 Task 8** [55]. It is a widely adopted benchmark dataset for relation extraction, specifically designed for multi-way classification of semantic relations between pre-identified entities. Each instance in the dataset is annotated with two entities and the semantic relation between them that requires classification. We use the dataset in the experiments exploring the cross-model transferability of TOA on the RE task (§4.2). Its training set contains 8,000 sentences, and the test set contains 2,717 sentences.

**CoNLL2000** [56]. It serves as a widely adopted benchmark for POS tagging and text chunking tasks. We use the dataset in the experiments exploring the cross-model transferability of TOA on the POS task (§4.2). Its training set contains 8,937 sentences, and the test set contains 2,013 sentences.

**IMDB** [57]. It is a widely used dataset for binary sentiment classification, containing movie reviews as its corpus. In the controlled experiments of the quantitative analysis for OLAS (§H), we utilized the corpus from this dataset. Both its training and test sets contain 25,000 sentences each.

# F  Implementation Details of Quantitative Analysis of OLAS

We sequentially sampled 1000 sentences from the CoNLL-2012 [54] dataset, i.e., $M$ in Equation 4 is 1000. Details about the dataset are in §E. We focus on the case where the source and target LMs do not belong to the same series, i.e., $S$ in Equation 4 is 4. For example, we use models of different sizes from the Llama and Qwen series as the source models, and models from the Gemma series as the target models. In this setup, there are significant differences in the research institutions, training data, model architectures, and tokenizers of the source and target models. To mitigate the effects of sentence length bias, we retained only the first 50 words of each sentence, including punctuation marks. Each sentence was assigned a unique identifier ranging from 1 to 1000, which was used to train the image classification model. For the image classification model, we utilized ResNet-18 [50] with the input channels modified to 1 and the output layer dimension set to 1000. Due to a small number of outliers that deviate noticeably from the overall distribution in both OLA and baseline results, the overall feature distribution is not clearly visible, as shown in Figure 3 (a), (b), and (d). To address this, we apply a consistent preprocessing procedure. Specifically, we calculate the mean $\mu$ and standard deviation $\sigma$ of each row, set outliers greater than $\mu + 3\sigma$ to zero, and then normalize the values by dividing by the row sum. Since the number of tokens obtained from sentence tokenization may not exactly correspond to the number of words in the original sentence, we resize all OLA maps to a $50 \times 50$ dimension. Additionally, as the OLA of CLMs is a lower triangular matrix, resizing introduces non-zero values in the upper triangular area. To avoid leaking tokenized sentence length information, we multiply the OLA by a lower triangular mask. Moreover, we employed common data augmentation techniques, including Gaussian noise, temperature scaling perturbations, and random highlighting. For each experimental setup, we trained with three learning rates: 1e-2, 5e-3, and 3e-3, and report the best-performing results. All experiments were conducted on a single 40GB NVIDIA A100 GPU.

---

[14]https://universaldependencies.org/

# G   Implementation Details of Transferable OLA Adapter

In this section, we present the implementation details of TOA for the four tasks: RE, NER, DP, and POS (§4.2). Since there are overlaps in the implementation details across these tasks, we consolidate their common aspects in the subsection below titled "Common Implementation Details". Additionally, due to the significant overlap between the experiments investigating the relationship between OLA and syntactic knowledge (§3.4) and the implementation of TOA on the DP task, these are discussed within the DP task section.

## G.1   Common Implementation Details

**OLA Preprocessing.** We apply the preprocessing operations described in §F to remove outliers and normalize both the OLA and baseline outputs. The processed results are then concatenated along the channel dimension to form the final input tensor. This tensor undergoes data augmentation using the operations described in §F.

**Feature Extractor in the Adapter.** Considering that the semantics of attention are distributed across rows and columns, where the $i$-th row represents the weights assigned to other tokens when the $i$-th token, as a query, aggregates other tokens, and the $i$-th column represents the weights assigned to the $i$-th token as a key when it is aggregated by other tokens, we use an axial transformer [58], which can extract semantics across rows and columns, to compose the feature extractor in our OLA adapter. The specific architecture of the feature extractor based on the axial transformer is as follows: the input map $X \in \mathbb{R}^{C \times L \times L}$ is first passed through a convolutional layer with a kernel size of $1 \times 1$ to increase its dimensionality to 768. It is then processed by several layers of axial transformers to produce the feature map $F_m \in \mathbb{R}^{768 \times L \times L}$, where $C$ represents the number of channels and $L$ represents the number of tokens. The diagonal features of are extracted to form the feature sequence $F_l \in \mathbb{R}^{768 \times L}$.

**Hyperparameters.** For the CLMs experiments, the axial transformer consists of 5 layers, while for the MLMs experiments, it consists of 3 layers. The number of epochs is set to 15. For each experimental setup, we trained with three learning rates: 1e-4, 3e-5, and 1e-5, and report the best-performing results.

## G.2   Task-Specific Implementation Details

**RE.** This task requires predicting the relationship between two entities in a sentence. Therefore, the adapter is structured to extract features of entity 1 and entity 2 from the feature extractor's output $F_l$ based on their annotated positions, concatenate these features, feed them into a fully connected layer, and produce the output $y \in \mathbb{R}^{19}$, where 19 denotes the number of relationship categories. The evaluation metric is the relationship classification accuracy.

**NER.** This task requires identifying entities in a sentence, where we convert the NER task into a sequence labeling task using BIO tagging [59]. Therefore, the adapter is structured to pass the feature extractor's output $F_l$ through a fully connected layer, producing the output $y \in \mathbb{R}^{37 \times L}$, where 37 denotes the number of BIO tag categories. The evaluation metric is the F1 score.

**DP.** This task requires identifying the head (governor) of each word in a sentence and its corresponding dependency relation. Therefore, the adapter is structured to pass the feature extractor's output $F_l$ through two separate MLP+biaffine modules [60], producing two outputs: $y_1 \in \mathbb{R}^{L \times L}$ for predicting the head of each token, and $y_2 \in \mathbb{R}^{55 \times L \times L}$ for predicting the dependency relation between each token and its head. Here, 55 denotes the number of dependency relation categories. The evaluation metrics are UAS (Unlabeled Attachment Score, measuring accuracy of head prediction without relation labels) and LAS (Labeled Attachment Score, measuring accuracy of both head and relation prediction).

**POS.** This task requires identifying the POS for each word in a sentence. Therefore, the adapter is structured to pass the feature extractor's output $F_l$ through a fully connected layer, producing the output $y \in \mathbb{R}^{45 \times L}$, where 45 denotes the number of POS categories. The evaluation metric is the POS classification accuracy.

# H Experimental Results with Controlled Dataset and Preprocessing

**Controlled Dataset Experiment.** When validating OLAS in §3.3.1, we sequentially selected the first 1,000 sentences of length 50 from CoNLL-2012. This approach avoids subjective bias through manual selection and ensures reproducible stability compared to random sampling. To verify whether the data selection generalizes to broader scenarios, we extended the experiments presented in Table 1 with the following three settings: (1) Replacing CoNLL-2012 with the IMDB dataset [57], (2) Increasing the sentence count from 1,000 to 2,000, and (3) Extending the sentence length from 50 to 80 words. As shown in Table 8, while experimental results vary across different data settings—primarily due to inherent differences in data similarity between datasets—**the consistently high performance under all configurations further validates the OLAS findings reported in the main text, demonstrating their reproducibility across diverse data settings.**

**Data Augmentation Ablation Study.** Our data preprocessing pipeline consists of three main steps: (1) Outlier Removal: Remove data points beyond three standard from row means. As shown in Figure 3, this prevents attention map suppression by extreme values in both OLA and baselines, ensuring stable training of the visual model. (2) OLA Size Standardization: Unify OLA dimensions across text sources to eliminate size-related information leakage, forcing the visual model to rely solely on visual features. (3) Data Augmentation: In our qualitative analysis, OLA generated by different LMs for the same text are grouped into one class for training the image classifier, resulting in only four samples per class (equal to the number of source models). Common augmentation are applied to alleviate overfitting in the visual model. Both outlier removal and OLA size standardization are fair and necessary operations. To verify whether the optional data augmentation introduces unintended bias, we conducted an ablation study (Table 9). Comparisons with Table 1 in the main text reveal that **while augmentation moderately reduces overfitting and improves classifier accuracy, removing it does not alter our core conclusion—OLAs still exhibit the most pronounced similarity patterns.**

Table 8: Quantitative OLAS analysis under three data settings. Rows 4–6: Results of dataset substitution (CoNLL-2012 → IMDb); Rows 8–10: Sentence count adjustments (1k → 2k); Rows 12–14: Sentence length extensions (50 → 80 words).

| Source | L-3b, L-8b, G-2b, G-9b | | L-3b, L-8b, Q-1b5, Q-7b | | Q-1b5, Q-7b, G-2b, G-9b | |
|---|---|---|---|---|---|---|
| Target | Q-1b5 | Q-7b | G-2b | G-9b | L-3b | L-8b |
| Dataset | | | | | | |
| 1st | 80.00 | 79.50 | 84.00 | 84.10 | 97.00 | 96.20 |
| 2nd | 70.60 | 65.00 | 77.90 | 73.00 | 91.50 | 92.60 |
| 3rd | 66.90 | 57.30 | 73.10 | 68.70 | 89.20 | 88.30 |
| Data Num | | | | | | |
| 1st | 72.80 | 64.35 | 86.25 | 83.05 | 93.55 | 94.15 |
| 2nd | 61.00 | 45.40 | 79.40 | 77.30 | 88.05 | 87.25 |
| 3rd | 63.10 | 33.85 | 77.05 | 74.40 | 85.50 | 84.95 |
| Data Len | | | | | | |
| 1st | 62.60 | 60.30 | 91.20 | 87.90 | 90.10 | 87.20 |
| 2nd | 58.60 | 42.20 | 97.70 | 96.60 | 95.00 | 93.80 |
| 3rd | 53.80 | 25.50 | 85.70 | 84.90 | 92.80 | 92.00 |

Table 9: OLAS Quantitative Evaluation Results Without Data Augmentation.

| Source | L-3b, L-8b, G-2b, G-9b | | L-3b, L-8b, Q-1b5, Q-7b | | Q-1b5, Q-7b, G-2b, G-9b | |
|---|---|---|---|---|---|---|
| Target | Q-1b5 | Q-7b | G-2b | G-9b | L-3b | L-8b |
| Rollout | 20.30 | 7.10 | 31.30 | 16.10 | 48.20 | 37.30 |
| IRNL | 8.90 | 5.00 | 47.10 | 57.50 | 70.60 | 64.80 |
| ALTI | 4.60 | 3.20 | 55.60 | 56.90 | 64.00 | 68.20 |
| 1st | 41.00 | 40.00 | 86.20 | 60.20 | **91.80** | **89.60** |
| 2nd | 50.30 | 40.80 | **86.50** | **76.80** | 85.90 | 83.90 |
| 3rd | **59.30** | **44.20** | 77.40 | 75.20 | 86.20 | 81.90 |

# I Experimental Results with Controlled Parameters

To ensure that the results of our quantitative analysis experiments (§3.3.1) reflect the inherent properties of LMs (i.e., the commonality of contextual aggregation patterns learned from large corpora) rather than confounds caused by data or other experimental biases, we conducted the following two controlled experiments:

- **Perturbations on Source Model Parameters**: We perturbed the parameters of the source LMs while keeping the target LM unchanged and repeated the OLAS quantitative analysis experiment (Table 1). The perturbations included two types: Random (randomizing model parameters) and Disorder (shuffling the order of model layers). As shown in Table 10, nearly no OLA similarity was observed between perturbed and normal models. **This indicates that the OLAS phenomenon is intrinsically tied to pre-trained model parameters, rather than arising from other experimental biases.**

- **Exploring OLAS Across Structurally Identical Models with Varied Parameters**: Taking Qwen2-1.5B as an example, the source models included four perturbed variants: Q-r1, Q-r2 (randomized parameters under different random seeds), and Q-d1, Q-d2 (disordered layers under different random seeds). The target models included the unperturbed Qwen2-1.5B and additional perturbed variants (Q-d3 and Q-r3). Under the configuration of the source and target LMs, we conducted the OLAS quantitative analysis experiments similar to those presented in Table 1. Because the source and target models share identical architectures, this stricter experimental setup isolates the impact of parameter variations. We further extended this analysis to Gemma2-9B and LLaMA3.2-3B. As shown in Table 11, no OLA similarity was observed between source and target models in any scenario, **further supporting the conclusion that OLAS depends on pre-trained parameters and eliminating confounds from other experimental setups.**

Table 10: Results of Perturbations on Source Model Parameters. Rows 4–6: Random (parameter randomization); Rows 8–10: Disorder (layer shuffling).

| Source | L-3b, L-8b, G-2b, G-9b | | L-3b, L-8b, Q-1b5, Q-7b | | Q-1b5, Q-7b, G-2b, G-9b | |
|---|---|---|---|---|---|---|
| Target | Q-1b5 | Q-7b | G-2b | G-9b | L-3b | L-8b |
| Random | | | | | | |
| 1st | 0.90 | 0.90 | 0.30 | 0.20 | 0.90 | 0.50 |
| 2nd | 1.20 | 1.00 | 0.90 | 1.60 | 0.40 | 1.30 |
| 3rd | 1.30 | 0.90 | 1.00 | 1.40 | 0.90 | 1.10 |
| Disorder | | | | | | |
| 1st | 0.70 | 0.60 | 1.20 | 1.30 | 1.10 | 0.80 |
| 2nd | 1.30 | 0.60 | 1.20 | 1.40 | 0.90 | 1.40 |
| 3rd | 0.80 | 1.00 | 0.90 | 1.50 | 1.10 | 1.60 |

Table 11: Results of exploring OLAS across structurally identical models with varied parameters.

| Source | Q-r1, Q-r2, Q-d1, Q-d2 | | | G-r1, G-r2, G-d1, G-d2 | | | L-r1, L-r2, L-d1, L-d2 | | |
|---|---|---|---|---|---|---|---|---|---|
| Target | Q-1b5 | Q-r3 | Q-d3 | G-2b | G-r3 | G-d3 | L-3b | L-r3 | L-d3 |
| 1st | 0.60 | 1.90 | 1.30 | 0.30 | 1.80 | 3.80 | 0.40 | 1.20 | 2.50 |
| 2nd | 1.50 | 3.10 | 1.30 | 1.80 | 1.40 | 3.60 | 0.40 | 2.50 | 2.70 |
| 3rd | 2.70 | 1.80 | 2.80 | 1.20 | 1.80 | 3.90 | 0.10 | 3.40 | 3.30 |

# J   Additional Result of Quantitative Analysis Based on Image Similarity Retrieval

We conducted retrieval-based quantitative evaluations on the cross-model similarity of OLA across first-, second-, and third-order configurations, with the results presented in Table 12.

Table 12: Retrieval-based quantitative evaluation of first- to third-order OLA cross-model similarity. Rows denote source LMs, columns represent target LMs, with entries reporting Hits@1 / Hits@5 metrics.

(a) CLM Results.

| Src\Tgt | Q-1b5 | G-2b | L-3b |
|---|---|---|---|
| | | 1st | |
| Q-1b5 | - | 83.60 / 89.40 | 95.90 / 97.00 |
| G-2b | 83.20 / 89.30 | - | 95.30 / 97.10 |
| L-3b | 92.90 / 96.10 | 94.10 / 96.50 | - |
| | | 2nd | |
| Q-1b5 | - | 75.20 / 83.60 | 91.50 / 95.60 |
| G-2b | 74.20 / 82.50 | - | 93.60 / 96.40 |
| L-3b | 85.40 / 92.60 | 91.80 / 95.20 | - |
| | | 3rd | |
| Q-1b5 | - | 57.10 / 72.00 | 69.90 / 88.40 |
| G-2b | 58.30 / 71.90 | - | 89.90 / 94.60 |
| L-3b | 64.40 / 81.60 | 86.40 / 92.90 | - |

(b) MLM Results.

| Src\Tgt | B-b | R-b | E-b |
|---|---|---|---|
| | | 1st | |
| B-b | - | 51.90 / 58.80 | 91.60 / 94.90 |
| R-b | 75.90 / 83.90 | - | 71.70 / 80.20 |
| E-b | 92.40 / 96.00 | 67.40 / 72.90 | - |
| | | 2nd | |
| B-b | - | 40.10 / 48.00 | 92.20 / 95.00 |
| R-b | 51.20 / 68.20 | - | 56.50 / 68.80 |
| E-b | 88.10 / 93.50 | 45.00 / 53.90 | - |
| | | 3rd | |
| B-b | - | 25.60 / 36.00 | 85.70 / 93.10 |
| R-b | 48.10 / 63.90 | - | 53.80 / 66.80 |
| E-b | 85.70 / 93.00 | 26.90 / 40.10 | - |

# K   Implementation Details of LLM Zero-Shot Evaluation

We use hand-crafted prompts to guide LMs in generating formatted outputs, then structure these outputs according to rules to extract their predictions. The prompt templates for RE and NER tasks are presented in Table 13, while those for DP and POS tasks are shown in Table 14.

Table 13: Prompt templates for RE and NER tasks.

| Task | Model type | Prompt template |
|------|-----------|-----------------|
| RE | MLM | Act as a relation extraction tagging tool. Find the relationship between e1 and e2 in the given sentence by choosing the correct option number from {REL_LABELS_STR}.

Sentence: {sentence}.
e1: {e1}
e2: {e2}
Response: The relationship number is {mask_str}. |
| | CLM | Act as a relation extraction tagging tool. Find the relationship between e1 and e2 in the given sentence according to these rules:
1. Choose the correct option number from {REL_LABELS_STR}.
2. Do not explain or add extra text. Only provide the option number.

Sentence: {sentence}.
e1: {e1}
e2: {e2}
Response: |
| NER | MLM | Act as a named entity recognition tagging tool. Given the sentence: "{sentence}", determine whether the span "{span}" is a named entity.
If not a named entity, respond strictly with "none".
If it is a named entity, select the correct category from {NER_LABELS_STR1} and respond only with the corresponding number.

Response: {mask_str}. |
| | CLM | Act as a named entity recognition tagging tool. Find all entities and their classes in a sentence according to these rules:
1. Choose the correct named entity class from {NER_LABELS_STR2}.
2. Do not explain or add extra text.

Sentence: {sentence}.
Response as tuples, and each tuple must have exactly two elements: first element is the named entity text (as a string), second element is the named entity class (as a string), e.g. (<entity1>, <class1>), (<entity2>, <class2>), ...
Response: |

Table 14: Prompt templates for DP and POS tasks.

| Task | Model type | Prompt template |
|------|-----------|-----------------|
| DP | MLM | Act as a dependency relation analyzing tool. Find the head and dependency relation of the given word in a sentence according to these rules:
1. Choose the correct head number from {words_map}.
2. Choose the correct dependency relation from {REL_LABELS_STR}.

Sentence: {" ".join(words)}
Word: {word}
Response: the head number is {mask_str}, the dependency relation is {mask_str}. |
| | CLM | Act as a dependency relation analyzing tool. Find the head and dependency relation of the
given word in a sentence according to these rules:
1. Choose the correct head number from {words_map}.
2. Choose the correct dependency relation from {REL_LABELS_STR}.
3. Do not explain or add extra text.

Sentence: {" ".join(words)}
Word: {word}
Response as a tuple which has exactly two elements: first element is the head number (as a int), second element is the dependency relation (as a str), e.g. (<head>, <relation>)
Response: |
| POS | MLM | Act as a part-of-speech (POS) tagging tool. Find the POS tag number of the given word in the given sentence by choosing the correct option number from {POS_LABELS_STR}.

Sentence: {sentence}.
Word: {word}.
Response: The POS tag number is {mask_str}. |
| | CLM | Act as a part-of-speech (POS) tagging tool. Find the POS tag of the given word in the given sentence according to these rules:
1. Choose the correct option number from {POS_LABELS_STR}.
2. Do not explain or add extra text. Only provide the option number.

Sentence: {sentence}.
Word: {word}
Response: |

## L  Additional Experimental Result of TOA

We investigate the cross-model generalization ability of TOA on the DP task and POS task. The experimental results are presented in Table 15 and Table 16. The comprehensive experimental results further support our conclusions in §4.2.

Table 15: Cross-Model Transferability of TOA on the DP Task. Entries represent UAS/LAS score (unit: %). Best performance is **bolded**; scores exceeding the zero-shot baseline are underlined.

(a) CLM Results.

| Src\Tgt | Q-1b5 | Q-7b | G-2b | G-9b | L-3b | L-8b |
|---|---|---|---|---|---|---|
| | Zero-shot | | | | | |
| - | 6.17/0.43 | 8.66/2.57 | 7.36/2.41 | 8.24/2.97 | 7.38/1.15 | 8.79/3.38 |
| | TOA (Ours) | | | | | |
| Q-1b5 | **65.44/50.48** | 45.96/27.32 | 45.32/26.83 | 38.97/18.77 | 49.02/31.11 | 45.24/26.70 |
| Q-7b | 52.63/33.69 | **61.93/47.24** | 35.94/17.44 | 31.45/13.74 | 45.87/24.88 | 47.95/26.93 |
| G-2b | 40.00/21.61 | 32.47/14.85 | **62.53/46.13** | 50.27/31.54 | 48.77/30.54 | 43.36/24.62 |
| G-9b | 36.93/16.81 | 30.48/12.61 | 51.62/32.75 | **61.05/44.27** | 44.49/24.78 | 40.18/20.35 |
| L-3b | 48.02/29.54 | 43.82/23.83 | 50.22/31.61 | 40.82/19.83 | **62.95/47.40** | 57.46/40.48 |
| L-8b | 48.44/30.35 | 47.03/27.83 | 46.87/28.05 | 35.08/16.23 | 58.53/42.86 | **60.99/45.77** |

(b) MLM Results.

| Src\Tgt | B-b | B-l | R-b | R-l | E-b | E-l |
|---|---|---|---|---|---|---|
| | Zero-shot | | | | | |
| - | 0.47/0.00 | 0.60/0.00 | 0.86/0.00 | 1.65/0.00 | 4.33/0.00 | 3.82/0.00 |
| | TOA (Ours) | | | | | |
| B-b | **81.15/72.07** | 71.19/57.62 | 58.02/38.10 | 58.14/37.98 | 71.11/55.90 | 71.02/55.28 |
| B-l | 67.38/52.89 | **80.95/71.15** | 43.77/20.54 | 54.31/32.38 | 59.05/37.95 | 62.52/41.87 |
| R-b | 65.70/51.30 | 52.50/36.55 | **78.35/67.51** | 63.90/45.64 | 68.49/54.21 | 69.64/55.15 |
| R-l | 64.58/49.80 | 58.91/43.45 | 64.11/48.20 | **78.92/68.23** | 59.01/42.04 | 55.65/37.11 |
| E-b | 72.15/58.83 | 62.84/46.78 | 62.52/45.14 | 56.66/36.49 | **80.33/70.93** | 76.84/65.45 |
| E-l | 68.88/55.76 | 63.26/47.87 | 64.45/48.27 | 53.35/33.46 | 73.92/61.98 | **80.77/71.24** |

Table 16: Cross-Model Transferability of TOA on the POS Task. Entries represent accuracy (unit: %). Best performance is **bolded**; scores exceeding the zero-shot baseline are underlined.

(a) CLM Results.

| Src\Tgt | Q-1b5 | Q-7b | G-2b | G-9b | L-3b | L-8b |
|---|---|---|---|---|---|---|
| | Zero-shot | | | | | |
| - | 3.88 | 22.76 | 7.77 | 59.77 | 22.05 | 39.00 |
| | TOA (Ours) | | | | | |
| Q-1b5 | **74.30** | 54.63 | 50.94 | 34.83 | 53.82 | 52.21 |
| Q-7b | 60.13 | **73.49** | 42.80 | 31.62 | 47.46 | 50.83 |
| G-2b | 53.03 | 41.79 | **73.78** | 57.45 | 51.52 | 47.92 |
| G-9b | 43.81 | 39.22 | 62.90 | **73.02** | 48.62 | 46.64 |
| L-3b | 56.96 | 50.48 | 57.18 | 45.09 | **73.20** | 68.35 |
| L-8b | 54.89 | 53.42 | 51.37 | 41.05 | 67.85 | **70.99** |

(b) MLM Results.

| Src\Tgt | B-b | B-l | R-b | R-l | E-b | E-l |
|---|---|---|---|---|---|---|
| | Zero-shot | | | | | |
| - | 0.44 | 0.66 | 0.23 | 0.65 | 0.67 | 0.60 |
| | TOA (Ours) | | | | | |
| B-b | **85.29** | 67.91 | 57.46 | 40.70 | 65.10 | 68.70 |
| B-l | 61.84 | **83.37** | 37.86 | 37.62 | 49.55 | 47.16 |
| R-b | 58.31 | 52.13 | **83.35** | 60.63 | 54.16 | 58.20 |
| R-l | 51.40 | 50.12 | 66.92 | **80.31** | 47.81 | 51.58 |
| E-b | 66.56 | 53.13 | 54.36 | 37.03 | **84.31** | 74.53 |
| E-l | 62.47 | 55.61 | 54.26 | 38.64 | 69.65 | **85.28** |

To conduct a more comprehensive evaluation of TOA's performance, we compare it with few-shot prompting and Cross-Model Control (CMC) [44]. The few-shot templates were constructed by adding the first five samples from the training set as demonstrations to the zero-shot templates (§K). CMC trains a delta LM and integrates its output with that of a LLM to enhance the latter's performance. Although this delta LM shares the same input-output format as the LLM and does not process features (thus not qualifying as an adapter), it has demonstrated considerable capability in

cross-model knowledge transfer. Therefore, we include it in the comparison. For the experimental setup, we adopt the widely used NER task and CLMs for evaluation. The results are presented in Table 17.

Table 17: Comparative experimental results on the NER Task. Entries represent F1 score (unit: %). Best performance is **bolded**.

| Src\Tgt | Q-1b5 | Q-7b | G-2b | G-9b | L-3b | L-8b |
|---|---|---|---|---|---|---|
| Zero-shot | | | | | | |
| - | 5.35 | 28.21 | 1.45 | 53.82 | 13.24 | 22.12 |
| Few-shot | | | | | | |
| - | 9.21 | 29.33 | 7.10 | **54.23** | 15.31 | 24.71 |
| CMC | | | | | | |
| Q-1b5 | 8.99 | 21.09 | 15.90 | 44.78 | 14.96 | 19.99 |
| G-2b | 1.03 | 7.19 | 29.78 | 39.96 | 1.71 | 1.93 |
| L-3b | 13.79 | 22.13 | 18.42 | 42.18 | 18.51 | 23.51 |
| TOA (Ours) | | | | | | |
| Q-1b5 | **53.81** | **30.99** | 21.08 | 9.58 | 29.24 | 26.94 |
| G-2b | 23.12 | 12.47 | **54.53** | 24.31 | 22.64 | 15.80 |
| L-3b | 27.48 | 21.79 | 24.79 | 14.05 | **54.51** | **45.24** |

To better analyze the reasons why LLMs underperform on fundamental NLP tasks, we have showcased several cases below using the NER task as an example. For Case 1, we show the full prompt (including the template from §K). For other cases, we display only the input sentence and model output; their prompts were generated by inserting the input after the template's 'Sentence:'.

Case1 (Q-7b):

```
Prompt: Act as a named entity recognition tagging tool. Find all entities
    and their classes in a sentence according to these rules: 1.Choose the
    correct named entity class from [PERSON, NORP, FACILITY, ORGANIZATION,
    Geo-Political Entity, LOCATION, PRODUCT, DATE, TIME, PERCENT, MONEY,
    QUANTITY, ORDINAL, CARDINAL, EVENT, WORK_OF_ART, LAW, LANGUAGE]. 2.Do
    not explain or add extra text. Sentence: Well , I especially want to
    know , ha , how the two of you found out the news on the day of the
    accident ? Response as tuples, and each tuple must have exactly two
    elements: first element is the named entity text (as a string), second
    element is the named entity class (as a string), e.g. (<entity1>,
    <class1>), (<entity2>, <class2>), ... Response:
Sentence: Well , I especially want to know , ha , how the two of you found
    out the news on the day of the accident ?.
Lables: (two, CARDINAL)
Outputs: (I, PERSON), (two of you, GROUP), (the news, PRODUCT), (the day,
    DATE), (the accident, EVENT)
```

Case2 (L-8b):

```
Sentence: From our perspective as media , I think it was precisely by going
    through those breaking - in , trials that we could be so calm in
    handling such an emergency today , ah , and give everyone the most
    effective announcements and notifications , and instructions .
Lables: (today, DATE)
Outputs: ('our perspective', 'EVENT') ('as media', 'FACILITY') ('I',
    'PERSON') ('it', 'PERSON') ('those breaking - in , trials', 'EVENT')
    ('today', 'DATE') ('ah', 'PERSON')
```

Case3 (Q-7b):

```
Sentence: Well , like Professor Zhou , I also received this news , ha ,
    through a mobile phone SMS .
Lables: (Zhou, PERSON)
Outputs: ('Professor Zhou', PERSON), ('news', EVENT), ('mobile phone SMS',
    PRODUCT)
```

Case4 (L-8b):

```
Sentence: I think , in comparison to China , we should say that urbanization
    in foreign countries developed earlier and is more widespread .
Lables: (China, Geo-Political Entity)
Outputs: ('China', 'Geo-Political Entity'),('urbanization',
    'EVENT'),('foreign countries', 'LOCATION')
```

**Analysis of the zero-shot and few-shot performance.** 1) Using NER as an example, LMs may not fully grasp certain nuances of the task. For instance (Cases 1 & 2), they often incorrectly predict pronouns like 'I' or 'you' as named entities. They also exhibit counterintuitive errors, such as labeling phrases like 'our perspective' as an 'EVENT'. 2) LLMs sometimes generate outputs that seem reasonable but diverge from dataset labels (Cases 3 & 4). This discrepancy resembles a mismatch with human labeling preferences. For example, where the dataset annotates 'Zhou' as 'Person', LMs prefer the full span 'Professor Zhou'. Similarly, LMs may classify 'mobile phone SMS' as 'PRODUCT', while the original labels do not. These represent comprehensible deviations from dataset conventions, rather than fundamental errors. 3) Few-shot prompting improves modestly over zero-shot, but gains remain limited. TOA typically outperforms both (Table 17). Smaller models (Qwen-1.5B, Gemma-2B, LLaMA-3B) benefit more from few-shot examples than larger counterparts (Qwen-7B, Gemma-9B, LLaMA-8B), likely because the limited information from only five examples adds negligible value to large models with extensive pre-trained knowledge.

**Analysis of CMC Results:** 1) CMC enhances an LM's performance by training a delta LM that concurrently reasons on the same input text alongside the source LM and integrates their output logits. Furthermore, this delta LM can be applied to enhance other target LMs, demonstrating promising results in instruction-following tasks. However, they utilized a delta LM (TinyLlama) and source LM (LLaMA2-7B) with identical tokenizers, and they only explored scenarios where the delta LM's tokenizer differed from the target LM's (e.g., LLaMA2-70B, Mistral-7B) via token mapping based on metrics like edit distance. We observed that when the source LM (e.g., Qwen-1.5B) and its delta LM (e.g., TinyLlama) employ different tokenizers, the knowledge transfer from the delta LM to target LMs faces challenges, leading to suboptimal performance. This performance gap likely arises from mismatches in tokenizer mapping between the source/target LMs and the delta LM, which can cause a distributional discrepancy in the tokens processed by the delta LM during both training and inference. Since the OLA of two LMs with different tokenizers remain similar, we did not encounter this problem. 2) We found that incorporating CMC may introduce previously absent formatting issues, such as nested parentheses, extra spaces, or superfluous commas. These artifacts somewhat lowered its overall score.

