# OpenReview forum: "Order-Level Attention Similarity Across Language Models: A Latent Commonality"
_NeurIPS.cc/2025/Conference — NeurIPS 2025 poster_

### Official Review · Reviewer_U4s6 · 2025-06-26

**Clarity:** 4
**Significance:** 4
**Originality:** 4
**Rating:** 5
**Confidence:** 4

**Summary:**

The paper presents a novel study exploring the commonalities in context aggregation patterns among LMs. First, it unifies the attention weights of the models into a comparable representation and introduces the concept of Order-Level Attention (OLA) derived from the decomposition of Attention Rollout. Both quantitative and qualitative methods reveal significant similarities in the same-order OLA across different LMs, and it is found that OLA encodes rich syntactic information. Based on this finding, the researchers propose the Transferable OLA Adapter (TOA), a training-free cross-model adapter transfer method. The experimental results demonstrate that TOA can effectively enhance the performance of LMs on syntactic-related tasks and facilitate cross-model knowledge transfer.

**Questions:**

1. What is the principle of linking OLA with syntactic knowledge?
2. Intuitively, the authors' findings stem from the observation that the OLA mappings of the same sentence in different models are highly similar. However, does this phenomenon also occur in sentences with similar semantic/syntactic structures? If semantically similar sentences do not show similar OLA mappings, but syntactically similar sentences do, then the motivation for this association may hold.
3. The TOA method shows ineffective results on G-9b, and I noticed that the OLA's ability to predict syntactic dependencies for G-9b is not reported in Table 3. Could there be different conclusions for G-9b? For example, does G-9b's OLA fail to accurately predict syntactic dependencies, or does it achieve better performance in predicting syntactic dependencies compared to other models? This could further explain the applicability of the TOA method.

**Ethical Concerns:**

["NO or VERY MINOR ethics concerns only"]

**Final Justification:**

The author effectively addresses my concerns about his motivations and analysis of the results. I hope this will be reflected in the new version.

**Limitations:**

yes

**Quality:**

4

**Strengths And Weaknesses:**

Strengths:

1. The research questions explored in this paper are far-reaching; exploring commonalities between different LMs can support cross-model knowledge transfer.
2. The research methods are comprehensive and insightful. From discovering common structural aspects to applications, the authors clearly explain the motivations and thought processes involved in the whole process, which will benefit similar studies.
3. This paper conducts extensive experimental verification and theoretical proof, which fully proves the rationality of the OLAS phenomenon.

Weaknesses:

1. The motivation for linking OLA with syntactic information is vague.


2. The applicability of the TOA method is limited.

On the one hand, TOA seems to focus mainly on syntax-related tasks, and on the other hand, it is only suitable for models that originally lack the ability in syntactic structure-related tasks. From the results of Gemma2-9B, TOA does not seem to achieve effective results when the model performs well. This may be because the improvement in TOA performance mainly comes from OLA, which contains rich general syntactic structure information. When the model already has strong capabilities in syntactic structure-related tasks, OLA's capabilities may not be enough to support its original capabilities, thus losing its advantages.

---

> ### Author Rebuttal · Authors · 2025-07-30
>
> Thank you for your insightful suggestions and positive feedback. Your input will help us refine this work, and comments such as 'The research questions are far-reaching' and 'The research methods are comprehensive and insightful' serve as strong encouragement to us.
>
> # 1. What is the principle of linking OLA with syntactic knowledge? (w1,q1,q2)
>
> Thank you for your insightful suggestions. Based on the two prompts below, we used GPT-4o to generate two sets of sentences. Group A contains 100 pairs of sentences with identical grammatical structures but opposite meanings. Group B contains 100 pairs of sentences with identical meanings but different grammatical structures. We used three CLMs to generate the OLA for these sentence pairs and computed their mean SSIM. As shown in Table 1 below, **sentence pairs with identical syntax but different semantics exhibit significantly higher similarity than pairs with identical semantics but different syntax. This further supports the view that OLA captures syntactic knowledge.**
>
> > 1. Prompt for generating Group A (syntactically identical, semantically opposite):
> > Please generate one hundred pairs of English sentences with identical grammatical structures but opposite meanings. The semantic opposition should be achieved by replacing one or more words in a sentence with words of the same part of speech but opposite meanings, resulting in sentences that convey contrary ideas. Each sentence must be exactly 50 words in length.
> > 2. Prompt for generating Group B (semantically identical, syntactically different):
> > Please generate one hundred pairs of English sentences with different grammatical structures but identical meanings. Each pair should convey the same semantic content, describing the same event or scenario, with each sentence strictly containing exactly 50 words.
>
> **Table 1: Image Similarity of OLA for Group A (100 sentence pairs with identical syntax/different semantics) and Group B (100 sentence pairs with identical semantics/different syntax). Values represent SSIM scores; higher values indicate greater image similarity (maximum = 1).**
> ||Q-1b5|G-2b|L-3b|
> |:-:|:-:|:-:|:-:|
> |Group A|0.8180|0.8346|0.8086|
> |Group B|0.6474|0.6497|0.6339|
>
> # 2. TOA’s Limitation in Applicability (w2)
>
> While OLA exhibits cross-model consistency, it has not yet achieved a perfect feature representation. It primarily captures syntactic information and encodes semantic information weakly. Thus, it is well-suited for syntax-related tasks but remains ineffective for semantics-heavy tasks (e.g., text generation). **We will continue refining it in subsequent work, aiming to uncover a shared semantic feature representation across LMs.**
>
> # 3. Does G-9b's OLA show significantly different performance in the dependency parsing task compared to other LMs? (q3)
>
> Due to space constraints, not all models' syntactic dependency parsing performances were included in Table 3 in the main text. Table 2 below shows the performance of Q-7b, G-9b, and L-8b's OLA on syntactic dependency parsing. It can be seen that **the performance of G-9b's OLA on this dependency parsing task is comparable to that of other LMs, showing neither unusually high nor low results.**
>
> The reason for TOA's weaker transfer performance on G-9b might lie in the similarity between G-9b's OLA and that of other models being slightly lower compared to other model combinations.
>
> **Table 2: Results of syntactic dependency parsing using OLA predicted by CLMs.**
> ||Q-7b|G-9b|L-8b|
> |:-:|:-:|:-:|:-:|
> |rollout|49.47/30.83|43.61/20.00|49.95/25.87|
> |1st|60.50/45.24|60.32/43.54|59.33/42.85|
> |2nd|56.74/40.36|56.00/37.96|54.25/36.81|
> |3rd|52.69/35.31|50.85/30.47|50.36/27.21|

---

> > ### Comment · Reviewer_U4s6 · 2025-08-05
> >
> > I have carefully read the author's response and can summarize it as follows:
> > 1. The authors conducted a targeted experiment to compare OLA similarity in syntactically similar vs. semantically similar sentence pairs. The results clearly show higher OLA similarity for syntactically similar pairs, thus substantiating the claim that OLA predominantly encodes syntactic knowledge. This convincingly addresses my earlier concern about the motivation behind this linkage.
> >
> > 2. The clarification on TOA’s syntactic focus and limited applicability to semantic tasks demonstrates a clear understanding of its scope. This limitation should be explicitly discussed in the paper.
> >
> > 3. The authors show that G-9b’s OLA achieves comparable parsing performance, and their hypothesis linking weaker transfer to lower OLA similarity across models is may reasonable. A deeper analysis of this case would further strengthen the work.

---

> > > ### Author Response · Authors · 2025-08-06
> > > **Appreciation for Your Insightful Review and Further Analysis of TOA Performance on Gemma-9b**
> > >
> > > **We sincerely appreciate the time and effort you dedicated to evaluating our work**, and are grateful for your constructive suggestions, which have been invaluable in guiding our revisions.
> > >
> > > Regarding your valuable suggestions:
> > > 1. **We are pleased that our targeted experiment addressed your earlier concern** and provided a clear explanation for the motivation linking OLA to syntactic knowledge.
> > > 2. Regarding TOA's scope limitation: We agree with your assessment that the limited applicability of TOA to semantic tasks requires explicit clarification. **In the revised manuscript, we will augment this discussion in Section 4.2.**
> > > 3. To further investigate the reasons for TOA's underperformance on G-9b, **we conducted two experiments to examine the OLA similarity between G-9b and other CLMs**:
> > >    1. **Using the image retrieval-based OLA similarity quantitative evaluation method (Section 3.3.2), we investigated the similarity between G-9b and other CLMs, with the results shown in ​Table 1​ below.** We found that ​G-9b achieved high retrieval success rates​ both as a source LM and as a target LM, although these values were somewhat lower than other model pairs (excluding the sibling model G-2b). **This indicates that ​the conclusion that 'OLA generated by different models for the same text are more similar than OLA generated for different texts' (line 205) still holds for G-9b**.​​
> > >    2. **In ​Table 2​ below, we report the average SSIM values of the OLA generated by G-9b and other CLMs ​for the same text.** Unlike Table 1, which reflects whether the OLA similarity for the same text is relatively higher than that for different texts, Table 2 reflects the ​absolute similarity​ of OLA generated for the same text by G-9b and other CLMs. We found that ​the SSIM values between G-9b and other CLMs (excluding G-2b) are lower than those between other model combinations.​​ For example, the SSIM between G-9b and Q-1b5 is only 0.76, whereas between G-2b and Q-1b5 it reaches 0.80. ​**This may explain the poor TOA performance on G-9b, namely a lower degree of OLA similarity between G-9b and other CLMs compared to that between other model combinations.**
> > >
> > > **Table 1: Retrieval-based quantitative evaluation of first-order OLA cross-model similarity. Settings identical to Table 2 in the main text.**
> > > |Src\Tgt|Q-1b5|G-2b|L-3b|G-9b|
> > > |:-:|:-:|:-:|:-:|:-:|
> > > |Q-1b5|-|83.60/89.40|95.90/97.00|83.20/88.80|
> > > |G-2b|83.20/89.30|-|95.30/97.10|100.00/100.00|
> > > |L-3b|92.90/96.10|94.10/96.50|-|93.40/95.60|
> > > |G-9b|81.10/87.60|100.00/100.00|94.50/96.20|-|
> > >
> > > **Table 2: Mean similarity of OLA for the same text under different LM combinations. Values in the table represent SSIM. Higher SSIM indicate greater similarity (maximum = 1).**
> > > |Src\Tgt|Q-1b5|G-2b|L-3b|G-9b|
> > > |:-:|:-:|:-:|:-:|:-:|
> > > |Q-1b5|-|0.8029|0.8315|0.7600|
> > > |G-2b|0.8029|-|0.8713|0.9350|
> > > |L-3b|0.8315|0.8713|-|0.8219|
> > > |G-9b|0.7600|0.9350|0.8219|-|

---

> > > > ### Comment · Reviewer_U4s6 · 2025-08-06
> > > >
> > > > Thank you for your further explanation. I think my concerns have been explained and the score has been raised to 5.

---

> > > > > ### Author Response · Authors · 2025-08-06
> > > > >
> > > > > Thank you again for your thorough review of our paper and for providing such thoughtful and enlightening feedback. We are truly honored to receive your recognition, which serves as a great encouragement to us.

---

### Official Review · Reviewer_dT4c · 2025-06-27

**Clarity:** 3
**Significance:** 2
**Originality:** 2
**Rating:** 4
**Confidence:** 5

**Summary:**

This paper investigates context aggregation patterns across LMs by using Order-Level Attention (OLA). The authors find that OLA patterns are consistent across models and align especially well with syntactic patterns. Leveraging this, they introduce a Transferable OLA Adapter (TOA) that shows some transfer abilities to unseen models without further training.

**Questions:**

- Do tokenizers need to be aligned for this to work? It appears that subtokens were aggregated in Figure 3. How was the aggregation performed?
- In Section 3.3, four LMs were using during training as the source LMs. Why did you decide to take several and how do the results depend on training on less source models, i.e. only one?
- “This suggests that OLA primarily encodes syntactic structures rather than semantic knowledge.” (l. 326) If I understand correctly, OLA maps do not directly encode representations but relationships of representations. This should be discussed more clearly. What are the limitations of this focus on relatedness over representations?

**Ethical Concerns:**

["NO or VERY MINOR ethics concerns only"]

**Final Justification:**

This was a good rebuttal and I appreciated the clarifications. The additional evaluations using Cosine Similarity and MSE are appreciated and confirm the reported benefits of their approach. The authors commit to completing the related work section on higher-order explanations to better contextualize their contribution in the broader research community. I would also suggest to clarify the focus on syntactic structural information, as this is crucial to consider for specific applications.

I have maintained my score of (4). I think overall this is a useful paper, but it may be a borderline accept as the originality and significance is limited. Overall, the paper appears ready for publication; if at NeurIPS remains to be seen.

**Limitations:**

Some specific limitations are discussed in Section 4.2 regarding the experimental results of the TOA experiments, but no general methodological limitations are discussed as part of the main paper.

**Paper Formatting Concerns:**

- Fontsize in Figure 3 should be increased, i.e. x/y-ticks and legend, I appreciate the text being repeated in the title though.

**Quality:**

2

**Strengths And Weaknesses:**

**Strengths**

- The authors address an interesting and relevant problem of (i) analyzing higher-order patterns in LMs and (ii) investigate good agreement of resulting patterns across different models.
- The methods are clearly defined and appear to be reproducible with reasonable efforts. Ablations, i.e. perturbation of the model parameters, further help to  support the results
- Extensive experiments on a variety of language models (CLMs, MLMs) and various tasks.

**Weaknesses**
- The main theoretical contribution appears to be a reformulation of attention rollout from a multi-order perspective.
- “We employed an image classification model as a proxy for human evaluation to objectively assess the similarity of OLA maps“  (l.182-183). I do not see how this is related to human evaluation. I was surprised to not see a more direct measure of relatedness, i.e. using correlation or distance measures between attention matrices (also in relation to the methods of representational alignment/representational similarity analysis). The image-based retrieval (Section 3.3.2 is a good step in this direction).
- This evaluation makes it difficult to infer the main result that “LMs exhibit significant similarity”, as it is accessed by a classification into the same group. It does not directly measure that the patterns across LMs are comparable or similar (as suggested in Figure 3). It could further be that different source models exhibit different patterns and the classification step merely matches it to one of these.
- The practical utility of the transfer setting (TOA) is not clearly demonstrated. While the authors show that training on the source model OLAs allows transfer to other LMs with improved performance over zero-shot, overall, performance is not competitive, e.g. for the NER task on CoNLL 2012 current state-of-the-art is at 80-84% with best scores on TOA is at around 69%.
- Missing related works on higher-order explanations that encode structural information such as messages between sequence elements (see relevant related works below).
- Lack of clear discussion of methodological limitations (see also questions below)


**Relevant related works:**

Higher-order explanations for DNNs:

- [Tsa18] M. Tsang, D. Cheng and Y. Liu, "Detecting statistical interactions from neural network weights", Proc. Int. Conf. Learn. Representations, 2018.
- [Ebe20] Eberle, O., Büttner, J., Kräutli, F., Müller, K. R., Valleriani, M., & Montavon, G. (2020). “Building and interpreting deep similarity models”, IEEE Transactions on Pattern Analysis and Machine Intelligence, 44(3), 1149-1161.
- [Jan20] J. D. Janizek, P. Sturmfels and S. Lee. "Explaining explanations: Axiomatic feature interactions for deep networks", CoRR, 2020.
- [Sch21] Schnake, T., Eberle, O., Lederer, J., Nakajima, S., Schütt, K. T., Müller, K. R., & Montavon, G. (2021). Higher-order explanations of graph neural networks via relevant walks. IEEE transactions on pattern analysis and machine intelligence, 44(11), 7581-7596.
- [Vas24] A Vasileiou and O Eberle. “Explaining Text Similarity in Transformer Models”, NAACL 2024.
- [Fum24] Fumagalli, F., Muschalik, M., Kolpaczki, P., Hüllermeier, E., & Hammer, B. (2024). KernelSHAP-IQ: Weighted Least-Square Optimization for Shapley Interactions. ICML 24.

---

> ### Author Rebuttal · Authors · 2025-07-30
>
> Thank you for your constructive suggestions and positive feedback. Your suggestions will help us refine this work, and comments such as 'addresses an interesting and relevant problem' and 'has clearly defined methods that appear reproducible with reasonable effort' serve as encouragement to us.
>
> # 1. OLA is a reformulation of attention rollout from a multi-order perspective.(w1)
>
> Although OLA is derived via order-level decomposition from Attention Rollout (Sec. 3.1), it differs from Attention Rollout and path tracing, which focus on quantifying input importance. Our focus, instead, is on uncovering common attention patterns across different LMs, an aspect not previously explored by these methods. **As demonstrated in Secs. 3.2 and 3.3, identifying cross-model attention commonalities proves difficult using Attention Rollout. In contrast, OLA, achieved through this straightforward decomposition, reveals significant similarities. This highlights the unique contribution of OLA from a novel perspective.**
>
> # 2. Evaluation of OLA Similarity. (w2, w3)
>
> **Regarding the question of whether the OLA generated by different LMs are comparable or similar, the answer is similar.** We visualized a large number of samples and observed that the OLA generated by different models for the same text do exhibit noticeable visual similarity (as shown in Fig. 3).
>
> For quantitative evaluation, we employed two methods: **one leveraging a visual model to assess the similarity of high-level features between different OLA**, and **the other evaluating low-level similarity using Image Similarity Retrieval** (Sec. 3.3).
>
> The rationale for introducing the visual model as a proxy for human evaluation is that it can assess the similarity of high-level features in images.
>
> To more directly demonstrate the cross-LM similarity of OLA, we conducted the following experiments:
>
> 1. We replaced the SSIM metric used in the OLA similarity evaluation experiment based on image retrieval (Sec. 3.3.2) with Cosine Similarity and Mean Squared Error (MSE). As shown in Tables 1 and 2 below, **using these simpler and more direct metrics, OLA still exhibit significant similarity.**
> 2. Using the data from Sec. 3.3.2, we calculated the similarity metrics (SSIM, Cosine Similarity, MSE) between OLA generated by the source LM and target LM for the same text input. We then computed the mean value across all texts. In simpler terms: For each of the 1000 texts, we generated an OLA pair (one from the source LM, one from the target LM), and we calculated the mean similarity metric across all 1000 OLA pairs. As shown in Table 4 below, **OLA generated for the same text by different LMs exhibit high similarity scores.**
>
> **Table 1: Retrieval-based quantitative evaluation of first-order OLA cross-model similarity. Settings identical to Table 2 in the main text. The similarity metric has been replaced with Cosine Similarity metric.**
> |Src\Tgt|Q-1b5|G-2b|L-3b|
> |:-:|:-:|:-:|:-:|
> |Q-1b5|-|83.40/88.70|94.20/96.80|
> |G-2b|78.70/85.80|-|97.40/98.60|
> |L-3b|90.60/93.90|96.70/97.80|-|
>
> **Table 2: Retrieval-based quantitative evaluation of first-order OLA cross-model similarity. The similarity metric is MSE.**
> |Src\Tgt|Q-1b5|G-2b|L-3b|
> |:-:|:-:|:-:|:-:|
> |Q-1b5|-|84.20/89.20|94.50/97.10|
> |G-2b|78.40/85.10|-|98.00/98.80|
> |L-3b|88.40/92.50|97.00/98.10|-|
>
> **Table 3: Mean similarity of OLA for the same text under different LM combinations. Values in the table represent (SSIM/Cosine Similarity/MSE). Higher SSIM and Cosine Similarity scores indicate greater similarity (maximum = 1). Lower MSE values indicate greater similarity (minimum = 0).**
> |Src\Tgt|Q-1b5|G-2b|L-3b|
> |:-:|:-:|:-:|:-:|
> |Q-1b5|-|0.8029/0.9605/0.00016|0.8315/0.9663/0.00013|
> |G-2b|0.8029/0.9605/0.00016|-|0.8713/0.9754/0.00010|
> |L-3b|0.8315/0.9663/0.00013|0.8713/0.9754/0.00010|-|
>
> # 3. The practical utility of TOA is not clearly demonstrated. (w4)
>
> TOA's performance has not yet reached the level of current state-of-the-art (SOTA) methods. As such, its practicality does have certain limitations. Nevertheless, other cross-model knowledge transfer approaches, such as the method in [1], face analogous challenges. **We maintain that this work represents a promising beginning and lays groundwork for future research.**
>
> [1] Zero-Shot Continuous Prompt Transfer: Generalizing Task Semantics Across Language Models.
>
> # 4. Related work is incomplete. (w5)
>
> Thank you for your feedback. **We will supplement these references in the subsequent version and include a discussion on them.**
>
> # 5. What are the limitations of this focus on relatedness over representations? (w6,q3)
>
> OLA fundamentally captures relationships between tokens, rather than token representations themselves. Our analysis (Sec. 3.4) demonstrates that OLA contains syntactic knowledge. We therefore utilize it as a specialized syntactic representation serving as input for TOA.  Leveraging OLA's cross-model similarity, TOA enables direct cross-model transfer. The limitation, however, is that compared to features generated by traditional learnable encoders, OLA suffers from information loss. **It primarily captures syntactic structural information, containing relatively limited semantic information. Consequently, this limitation restricts TOA's performance to semantics-dependent tasks.**
>
> # 6. Do tokenizers of different LMs require alignment? Should subtokens be aggregated? (q1)
>
> **Alignment of different LMs' tokenizers is not required, nor is aggregation necessary.**
>
> As evidenced in Fig. 3, despite Qwen and LLaMA having different tokenizers, their OLA for the same text remains similar.
>
> We did not aggregate subtokens. The axes in Fig. 3 display raw subtokens.
>
> # 7. Why does Sec. 3.3.1 use 4 source LMs instead of 1? (q2)
>
> Employing four source LMs rather than fewer is intended to mitigate overfitting in the visual model.
>
> We treat OLA generated from the same text by different LMs as belonging to the same class (line 183). This implies that the number of training samples per class equals the number of source LMs. **If only one source LM were used, each class would contain just a single sample. This would lead to overfitting in the visual model, preventing it from learning comprehensive features of OLA. Using four source LMs helps mitigate this issue.**
>
> Table 4 below presents the results of cross-model similarity evaluation for OLA based on the visual model when using only a single source LM (analogous to Table 1 in the main text), which validates the above perspective.
>
> **Table 4: Results of quantitative evaluation on cross-model similarity for OLA based on visual model using only a single source LM.**
> |src\tgt|Q-1b5|Q-7b|G-2b|G-9b|L-3b|L-8b|
> |:-:|:-:|:-:|:-:|:-:|:-:|:-:|
> |Q-1b5|-|95.70|19.10|6.60|8.10|9.80|
> |G-2b|9.30|8.90|-|74.60|39.70|36.50|
> |L-3b|10.50|7.70|48.00|41.40|-|96.50|

---

> > ### Comment · Reviewer_dT4c · 2025-08-05
> > **Rebuttal Response**
> >
> > Thank you for the honest and detailed response, this is a good rebuttal and I appreciate the clarifications.
> >
> > The additional evaluations using Cosine Similarity and MSE are appreciated and confirm the reported benefits of their approach.
> >
> > The authors commit to completing the related work section on higher-order explanations to better contextualize their contribution in the broader research community.
> >
> > I would also suggest to clarify the focus on syntactic structural information, as this is crucial to consider for specific applications.
> >
> > Overall, I think this is a good paper and appears ready for publication.

---

> ### Author Response · Authors · 2025-08-05
> **Appreciation for Your Insightful Review and Commitment to Improvements​**
>
> **Thank you for your insightful review and constructive feedback.** ​​We are honored that our rebuttal resonated with you and received your positive acknowledgment.​​ In subsequent versions, we will:
>
> 1. **Expand the Related Work​​ to thoroughly discuss higher-order explanations**, contextualizing our contribution.
> 2. ​**Clarify our focus​​ on syntactic structural information**, emphasizing its scope and application relevance.
>
> ​
> **Your affirmation that the paper is ‘ready for publication’ is tremendously encouraging.** We deeply value your support!

---

### Official Review · Reviewer_uyDY · 2025-07-03

**Clarity:** 4
**Significance:** 3
**Originality:** 3
**Rating:** 5
**Confidence:** 4

**Summary:**

From the perspective of information flow, this paper decomposes the information flow formed by multi‑head attention (MHA) and residual connections, proposing Order‑Level Attention (OLA). Through a series of qualitative and quantitative experiments, it demonstrates the validity of OLA, reveals its potential as an invariant shared across different large language models (LLMs), and then introduces Transferable OLA Adapter (TOA). Experiments on four basic NLP tasks further verify TOA’s effectiveness and, by extension, the soundness of OLA.

**Questions:**

1. Have you tried applying TOA to transfer for text‑generation tasks?

2. Why are there no experiments for fourth‑order OLA and above? Is it because higher‑order features are heavily affected by the attention‑sink phenomenon?

**Ethical Concerns:**

["NO or VERY MINOR ethics concerns only"]

**Final Justification:**

The authors have addressed my concern regarding whether the TOA method can be applied to text generation. I have decided to keep the score unchanged.

**Limitations:**

yes

**Quality:**

3

**Strengths And Weaknesses:**

> Strengths

- The paper offers a novel viewpoint by decomposing the information flow generated by MHA and residual connections, uncovering a common contextual aggregation pattern among LLMs and proposing OLA.

- Experiments are thorough and well designed. Qualitative studies first reveal the similarity of same‑order OLAs (OLAS) across different LLMs for the same text; this is then substantiated through carefully crafted image‑classification and image‑similarity‑retrieval experiments, as well as quantitative analyses of “Relation between OLA and Syntactic Knowledge.”

- The proposed methods are simple yet effective; on quantitative tasks based on image classification and image‑similarity retrieval they substantially outperform a range of baselines.

> Weaknesses

- The TOA method is restricted to syntax‑related tasks and cannot be directly applied to text‑generation or other semantics‑heavy tasks.

---

> ### Author Rebuttal · Authors · 2025-07-30
>
> Thank you for your valuable feedback and kind recognition of our work. Your comments such as 'The paper offers a novel viewpoint,' 'Experiments are thorough and well designed,' and 'The proposed methods are simple yet effective' have been extremely encouraging to us.
>
> # 1. TOA’s Limitation in Applicability. (w1,q1)
>
> While OLA exhibits cross-model consistency, it has not yet achieved a perfect feature representation. It primarily captures syntactic information and encodes semantic information weakly. Thus, it is well-suited for syntax-related tasks but remains ineffective for semantics-heavy tasks (e.g., text generation).
>
> We previously considered using OLA for text generation tasks and conducted exploratory experiments. When attempting to reconstruct original sentences using OLA, we observed unsatisfactory results. Consequently, we simplified the task by employing OLA for part-of-speech (POS) tagging of the original sentences, which yielded well-received outcomes. Since POS correlates strongly with syntactic structures, this realization directed our research toward syntax-oriented exploration. **Although OLA currently primarily encapsulates syntactic knowledge, we will continue pursuing a unified semantic feature representation across diverse LLMs.**
>
> # 2. Discussion on Higher-Order OLA (q2)
>
> In the main text, we primarily analyze 1st to 3rd-order OLA because they have already demonstrated significant cross-model similarity and possess lower computational complexity. The computational complexity of higher-order OLA is higher: for k-th order OLA, it is necessary to calculate $\binom{N}{k}$ paths worth of information aggregation matrices, where N denotes the number of LM layers. Therefore, we currently focus less on higher-order OLA.
>
> In Table 1 below, we present the quantitative analysis results for fourth-order OLA cross-model similarity (settings mirror main-text Table 2). **We discover that fourth-order OLA still reveals significant commonalities. In future work, we will explore higher-order OLA to formalize theoretical generalization boundaries.**
>
> **Table 1: Retrieval-based quantitative evaluation of fourth-order OLA cross-model similarity. Entries indicate evaluation metrics Hits@1 / Hits@5 (unit: %)**
> |Src\Tgt|Q-1b5|G-2b|L-3b|
> |:-:|:-:|:-:|:-:|
> |Q-1b5|-|48.50/69.60|58.80/80.30|
> |G-2b|45.90/66.00|-|89.30/94.00|
> |L-3b|53.70/76.80|88.70/93.30|-|

---

> > ### Comment · Reviewer_uyDY · 2025-08-05
> >
> > Thank you for the response. I have decided to keep the score unchanged.

---

> ### Author Response · Authors · 2025-08-05
> **Appreciation for Your Review and Continued Engagement​**
>
> **Thank you sincerely for taking the time to thoroughly review our manuscript and for sharing your thoughtful feedback.** ​​Your positive feedback is highly valued and encouraging for our research.​​

---

### Official Review · Reviewer_LYHK · 2025-07-05

**Clarity:** 3
**Significance:** 2
**Originality:** 3
**Rating:** 4
**Confidence:** 4

**Summary:**

This paper investigates whether different large language models (LLMs) exhibit common patterns in how they aggregate contextual information through attention mechanisms. The authors propose a novel attention decomposition method called Order-Level Attention (OLA), which breaks down cumulative attention rollout into multiple “orders” (e.g., 0th-order: residual path only, 1st-order: single-layer aggregation, and so on). They observe that OLA representations of the same order across different models show a high degree of similarity, a phenomenon they term Order-Level Attention Similarity (OLAS). Building on this insight, the authors introduce the Transferable OLA Adapter (TOA), a training-free adapter mechanism that enables cross-model transfer. Experiments conducted on 12 different models and multiple downstream tasks—including relation extraction, part-of-speech tagging, and named entity recognition—demonstrate the effectiveness of their approach.

**Questions:**

- The current experiments do not clearly establish the advantage boundaries of TOA. Could you add comparisons with other representation transfer methods, such as Representation Editing, Middle-layer Alignment, or Hidden-State Transfer? Also, how does TOA perform relative to trained adapters like LoRA , or Prefix Tuning? Is there a trade-off between accuracy and training cost?
- Have you evaluated OLA or TOA on models with significantly different architectures, such as encoder-only versus decoder-only Transformers? How well does the method generalize across these divergent settings?
- How does OLA behave in long-sequence settings? In such cases, what changes (if any) occur in TOA's training complexity, inference latency, and accuracy? Does high-order OLA risk overestimating distant token contributions in longer contexts?
- The current experiments focus exclusively on tasks suitable for encoder-only models (RE, NER, DP, POS). Can you expand the evaluation to include tasks like text continuation or summarization, which are more natural for causal language models (CLMs), to validate OLA and TOA in that setting?
- The paper reports the performance differences across different OLA orders (first-order, second-order, and third-order) in Table 2 (retrieval task) and Table 3 (dependency parsing task), with notable variations observed across both CLM and MLM model families. However, the authors do not provide sufficient analysis or discussion to explain these differences.  For example:
  - In Table 3(a), the Qwen2-1.5B model achieves a UAS/LAS of 63.58/48.24 using first-order OLA on the dependency parsing task, which is significantly higher than the 55.19/36.82 performance obtained with third-order OLA.
  - Moreover, in the ablation study on data augmentation in Appendix Table 9, the cross-model transfer results vary across different OLA orders, showing inconsistent trends across different models.
  - This raises the question of whether the choice of OLA order has a complex, model- and task-dependent impact on performance.

**Ethical Concerns:**

["NO or VERY MINOR ethics concerns only"]

**Final Justification:**

I have no further concerns and believe that 4 is an appropriate score for this paper.

**Limitations:**

OLA performs an order-wise decomposition of attention rollout. However, attention rollout itself has been primarily applied in encoder-based models such as BERT, where it serves to quantify input token importance across layers. Its applicability to causal language models (CLMs) remains unclear and is not addressed in the paper. In addition, although the paper presents a wide range of experiments, many of them remain relatively superficial. The baselines used are fairly basic, and several important issues in the experimental data are not thoroughly examined or discussed.

**Quality:**

3

**Strengths And Weaknesses:**

**Strengths**
- OLA introduces a novel perspective for unified attention representation by decomposing attention rollout into order levels from a path-based view, facilitating interpretability and structural alignment across different language models.
The authors measure attention structure similarity using image-based methods and support their OLAS observation with thorough theoretical analysis and empirical validation.
- TOA leverages attention representation for transfer and offers a simple, training-free mechanism that outperforms baseline models in multiple tasks.
- The paper covers both theoretical formulation and practical application, showing strong breadth and depth.

**Weaknesses**
- OLA bears substantial similarity to existing methods like Attention Rollout or path tracing (e.g., Abnar & Zuidema, 2020). Attention Rollout is already a widely used method for quantifying input importance at any layer of encoder-based models like BERT. Its applicability to causal language models (CLMs) remains under-discussed. Furthermore, the proposed order-wise decomposition may amplify the overestimation of distant token importance in long sequences—an issue not addressed in the paper.
- In the experiments, the authors did not adequately compare against other interpretability methods (e.g., LogitLens, ACDC) or transfer approaches (e.g., LoRA, P-Tuning). The main baseline used is zero-shot, which is a relatively weak comparison. No comparisons are made to representation transfer methods such as hidden-state alignment or DSKD, leaving the true advantage of TOA unclear.
- The claim of being the “first” to achieve training-free adapter transfer is overstated, as similar ideas have been explored in prior works.
Theoretical properties, generalization boundaries, and computational efficiency of high-order OLA are not clearly discussed.

---

> ### Author Rebuttal · Authors · 2025-07-30
>
> Thank you for your constructive suggestions and positive feedback. Your insights will help us improve the work, and comments such as 'a novel perspective' and 'strong breadth and depth' are a great encouragement.
>
> # 1. The Innovativeness of OLA and its Relation to Interpretability Methods (w1, w2)
>
> Although OLA derives from Attention Rollout via order-level decomposition (Sec. 3.1), it differs from Attention Rollout and path tracing, which focus on quantifying input importance. Instead, our focus is on uncovering common attention patterns across different LMs, a novel aspect unaddressed by these methods. As shown in Secs. 3.2 and 3.3, identifying these cross-model commonalities is difficult with Attention Rollout. **In contrast, OLA, achieved through this straightforward decomposition, reveals significant similarities, highlighting its unique value from a novel perspective.**
>
> Regarding the relationship to interpretability methods: OLA originates from Attention Rollout, an interpretability method, and thus retains the ability to explain model behavior. However, this work is not aimed at enhancing model interpretability per se. **Our primary goal is to discover latent commonalities in attention patterns across LMs. Consequently, we did not include interpretability methods like LogitLens or ACDC (which analyze individual model behavior and don’t study cross-model commonalities) in our comparative analysis.**
>
> # 2. Is there a trade-off between training cost and performance for TOA? (w1)
>
> Yes, **there is a trade-off between TOA's training cost and performance**. As shown in Table 1 below, performance increases with more training steps, peaking at ~149,004 steps (taking ~7 hours). Beyond this, gains slow and performance eventually plateaus.
>
> **Table 1:  TOA's F1 scores on NER with increasing training steps. (the source and target LM are Qwen2-1.5B)**
>
> |steps|49,668|99,336|149,004|198,672|248,340|
> |:-:|:-:|:-:|:-:|:-:|:-:|
> |F1|38.66|45.19|53.81|53.75|52.97|
>
> # 3. Applicability of OLA to CLMs and Long Sequences (w1,q3)
>
> As shown in Section 3, OLA reveals significant commonalities across both CLMs and MLMs. As demonstrated in Section 4, TOA achieves cross-model knowledge transfer on both CLMs and MLMs. **This proves OLA's applicability to both CLMs and MLMs.**
>
> For long sequences, we conducted two experiments:
>
> 1. **We validated OLA's cross-model similarity on long sequences.** We sampled 1000 sentences of length 256 from CoNLL-2012 and repeated the experiment corresponding to Table 2 in the main text. Results are shown in Table 2 below. Retrieval accuracy improves with longer sentences (vs. main text Table 2), because increased sentence length enhances structural diversity, making OLA patterns between different sentences more distinct, thereby boosting retrieval success.
> 2. **We validated TOA's transfer performance, training time, and inference speed on long sequences.** For the NER task (Section 4.2), we created length-256 sequences by randomly concatenating original sentences. Results are shown in Table 3 below. TOA's F1 score shows comparable performance vs. main text Table 5, indicating its suitability for long sequences. Regarding training complexity and inference latency, Table 4 shows both increase with sequence length. This issue can potentially be addressed in the future through architectural optimization of TOA.
>
> Regarding whether OLA overestimates the importance of distant tokens in long sequences: As observed in Fig. 3(b) and (d), OLA tends to overestimate the importance of distant tokens, particularly the first token, due to the attention sink phenomenon (Section 3.1). However, **this does not appear to affect OLA's core commonality findings or TOA's performance, since OLA patterns still exhibit high similarity across different models - including this shared tendency to overestimate distant token importance.**
>
> **Table 2: Retrieval-based quantitative evaluation of first-order OLA cross-model similarity using long-sequence.**
> |Src\Tgt|Q-1b5|G-2b|L-3b|
> |:-:|:-:|:-:|:-:|
> |Q-1b5|-|92.90/95.50|97.00/98.60|
> |G-2b|94.00/96.20|-|91.10/93.90|
> |L-3b|96.70/98.00|91.60/93.60|-|
>
> **Table 3: Cross-LM transferability of TOA on the NER task using long-sequence.**
> |Src\Tgt|Q-1b5|G-2b|L-3b|
> |:-:|:-:|:-:|:-:|
> |Q-1b5|55.84|19.18|25.37|
> |G-2b|27.40|55.81|19.31|
> |L-3b|36.26|31.13|52.11|
>
> **Table 4: TOA's training time and inference latency across sequence lengths on the NER task.**
> |length|training time (s/step)|inference latency (s)|
> |:-:|:-:|:-:|
> |50|0.17|0.01|
> |256|3.74|0.76|
>
>
> # 4. Supplemental Comparative Methods (w2,q1):
>
> Our approach focuses on training-free adapter transfer, operating without accessing the target model during transfer. **While the knowledge distillation methods you mentioned, such as hidden-state alignment [1] or DSKD [2], offer valuable insights (and will be discussed in future version), they require training student models, falling outside our training-free scope. Consequently, they were excluded from our baselines.**
>
> Regarding LoRA and P-tuning methods:
> 1. P-tuning: We adapted this as a baseline by directly transferring soft prompts trained on the source LM to target LMs. When encountering dimensional mismatches in word embeddings, we performed interpolation to resize prompts to the target LM's dimensions.
> 2. LoRA: It cannot be directly applied in cross-LM transfer due to architectural incompatibilities (e.g., differing layer counts and hidden dimensions between models).
>
> P-tuning results (Table 5 below) show that transferred soft prompts achieved near zero-shot performance on Qwen models but failed completely on Gemma and LLaMA. This indicates that:
> 1. This poor performance, approaching zero-shot levels or failing entirely, confirms P-tuning's unsuitability for cross-model transfer.
> 2. This discrepancy may stem from tokenizer differences: Qwen's tokenizer lacks a BOS token, making it robust to noisy soft prompts occupying the first token position. In contrast, Gemma and LLaMA require a BOS token as the first token to function correctly.
>
> **Table 5: Cross-Model Transferability of P-tuning.**
> |src\tgt|Q-1b5|Q-7b|G-2b|G-9b|L-3b|L-8b|
> |:-:|:-:|:-:|:-:|:-:|:-:|:-:|
> |Q-1b5|74.36|28.84|0.00|0.00|0.00|0.00|
> |G-2b|6.15|27.86|81.39|0.00|0.00|0.00|
> |L-3b|5.32|30.20|0.00|0.00|53.00|0.00|
>
> [1] IMPROVING LANGUAGE MODEL DISTILLATION THROUGH HIDDEN STATE MATCHING.
> [2] Dual-Space Knowledge Distillation for Large Language Models.
>
> # 5. Overstatement of 'first' (w3)
>
> Thank you for noting this. We are the first to identify the OLA similarity phenomenon. Regarding cross-model transfer, a limited number of studies have investigated this. **We will refine our expression.**
>
> # 6. Discussion on High-Order OLA (w3)
>
> **In the main text, we primarily analyze 1st to 3rd-order OLA because they have already demonstrated significant cross-model similarity and possess lower computational complexity.** The computational complexity of higher-order OLA is higher: for k-th order OLA, it is necessary to calculate $\binom{N}{k}$ paths worth of information aggregation matrices, where N denotes the number of LM layers. Therefore, we currently focus less on higher-order OLA.
>
> Table 6 below presents quantitative results for fourth-order OLA similarity (settings match main text Table 2). We discover that fourth-order OLA still reveals significant commonalities. In future work, we will explore higher-order OLA to formalize theoretical generalization boundaries.
>
> **Table 6: Fourth-order OLA cross-model similarity retrieval scores. Entries: Hits@1 / Hits@5 (%)**
> |Src\Tgt|Q-1b5|G-2b|L-3b|
> |:-:|:-:|:-:|:-:|
> |Q-1b5|-|48.50/69.60|58.80/80.30|
> |G-2b|45.90/66.00|-|89.30/94.00|
> |L-3b|53.70/76.80|88.70/93.30|-|
>
> # 7. OLA Similarity Analysis between CLM and MLM (q2)
>
> Table 7 below shows first-order OLA similarity between encoder-only (MLM) and decoder-only (CLM) Transformers, mirroring main text Table 2's settings. Since CLM OLA matrices are lower triangular but MLM ones are not, we masked the MLM matrices to lower triangular form for SSIM calculation.
>
> As shown, **OLA similarity between CLMs and MLMs is lower than within CLM or MLM pairs, but still present in some degree.** This indicates shared contextual information aggregation mechanisms across these architectures.
>
> **Table 7: Retrieval-based quantitative evaluation of first-order OLA between MLMs and CLMs.**
>
> |Src\Tgt|Q-1b5|G-2b|L-3b|
> |:-:|:-:|:-:|:-:|
> |B-b|12.40/26.90|12.40/24.30|11.30/26.90|
> |R-b|13.30/36.40|16.10/37.20|17.70/34.70|
> |E-b|6.60/20.20|3.80/13.00|6.70/16.80|
>
> # 8. TOA’s Limitation in Applicability (q4):
>
> As stated in Sec 4.2, while OLA exhibits cross-model consistency, it has not yet achieved a perfect feature representation. It primarily captures syntactic information and encodes semantic information weakly. Thus, it is well-suited for syntax-related tasks but remains ineffective for semantics-heavy tasks (e.g., text generation). **Developing cross-model unified semantic representations is a key direction for our future work.**
>
> # 9. Selection of the OLA Order (q5)
>
> The OLA similarity does vary to some extent for different model pairs and orders, but the overall trend shows high similarity from orders 1 to 3 across most pairs (line 217), proving that OLA similarity is a widespread phenomenon.
>
> To investigate how OLA order selection would affect TOA's performance, we tested TOA's performance under different orders for the NER Task. Results in Table 8 below show no substantial performance differences across orders 1 to 3. Therefore, **given that order-1 OLA has the lowest computational complexity, it is generally sufficient in most scenarios.**
>
> **Table 8: TOA Transfer Performance Under Different Orders. Source LM is Qwen2-1b5.**
> |Orders\Tgt|Q-1b5|Q-7b|G-2b|G-9b|L-3b|L-8b|
> |:-:|:-:|:-:|:-:|:-:|:-:|:-:|
> |1st|54.12|32.69|22.31|9.89|27.65|27.23|
> |2nd|52.79|31.23|20.88|9.67|24.12|20.92|
> |3rd|51.52|27.20|20.35|9.48|13.59|11.55|

---

> > ### Comment · Reviewer_LYHK · 2025-08-07
> >
> > Thank you for your response. After reading your rebuttal, I decide to maintain my score.

---

> > > ### Author Response · Authors · 2025-08-07
> > >
> > > Thank you once again for your thorough review of our paper and your insightful comments. Your positive feedback is both helpful and encouraging for our research.

---

### Official Review · Reviewer_aRHQ · 2025-07-05

**Clarity:** 2
**Significance:** 2
**Originality:** 2
**Rating:** 2
**Confidence:** 4

**Summary:**

This paper investigates the similarity of context aggregation patterns across different language models. The authors introduce Order-Level Attention (OLA) as a method to characterize these patterns. Specifically, the i-th order OLA is defined as the sum of the i-th order terms in the expansion of the attention rollout. The authors demonstrate that OLA exhibits notable similarities across models and is associated with syntactic knowledge. Building on these observations, they propose the Transferable OLA Adapter (TOA), claimed to be the first approach enabling cross-LM adapter transfer without additional tuning. By leveraging OLA as a unified syntactic representation across LMs, the adapters are trained with OLA for downstream tasks. Since OLA appears to be similar across models, the trained adapters can be directly applied to other LMs without further adaptation.

**Questions:**

1.	Can TOA generalize to more challenging tasks that require deeper semantic understanding, such as generative tasks like question answering?
2.	I am puzzled as to why 7B-scale instruct models (e.g., Qwen2-7B, LLaMA 3.1-8B) perform poorly on these tasks, given that models of this size have achieved strong results on more difficult mathematical benchmarks. Could the authors provide further analysis or case studies to explain this behavior? How does TOA performance compare to few-shot prompting?
3.	What is the rationale for employing visual models or image similarity metrics to assess similarity, rather than simpler measures such as vector similarity?

**Ethical Concerns:**

["NO or VERY MINOR ethics concerns only"]

**Limitations:**

yes

**Quality:**

2

**Strengths And Weaknesses:**

Strengths
1.	The proposed TOA enables transfer across different language models without requiring additional training.
2.	The authors employ a range of qualitative and quantitative analyses to examine OLA similarity across LMs.
3.	The experiments are conducted on models of varying sizes and architectures.

Weaknesses
1.	The evaluation is limited to NLP tasks involving syntactic and simple semantic understanding; more complex tasks are not considered.
2.	There is no comparison against related methods, such as Cross-model Control: Improving Multiple Large Language Models in One-time Training.
3.	The approach requires modifying the adapter structure and training separate adapters for different tasks, limiting its universality.
4.	The cross-model generalization capability appears limited; transferring to other models often results in performance degradation exceeding 10 percentage points.
5.	Some descriptions lack clarity. For example, Sec 3.2 refers to “zeroing maxima for first-/second-order OLA,” while Appendix B mentions “setting outlier values in the OLA to zero,” creating inconsistencies. Additionally, Section G.1 contains an incomplete phrase: “The diagonal features of …”

---

> ### Author Rebuttal · Authors · 2025-07-30
>
> We appreciate your insightful suggestions.
>
> # 1. TOA’s Limitation in Applicability (w1, q1):
>
> As stated in Sec 4.2, while OLA exhibits cross-model consistency, it has not yet achieved a perfect feature representation. **It primarily captures syntactic information and encodes semantic information weakly. Thus, it is well-suited for syntax-related tasks but remains ineffective for semantics-heavy tasks** (e.g., text generation).
>
> **Developing cross-model unified semantic representations is a key direction for our future work.** We're integrating the technique from [1] into OLA by representing each word's semantics as its relevance to a set of Anchor Words. This approach holds promise to enhance OLA's semantic capacity while preserving its cross-model consistency.
>
> Moreover, **a significant portion of our contribution is reflected in the discovery of OLAS**, which reveals universal attention patterns across diverse LMs, a finding acknowledged by other reviewers.
>
> # 2. Supplemental Comparative Methods and Further Performance Analysis for Zero-shot (w2, q2)
>
> In the subsequent version, **we will supplement comparative methods including Cross-Model Control (CMC) [2] and few-shot prompting.** Below we present the partial results for the NER task (Sec. 4.2). The few-shot templates were constructed by adding the first five samples from the training set as demonstrations to the zero-shot templates (Sec. K).
>
> **Table 1: Comparative experimental results (F1 score). Best performance bolded.**
> |method|src\tgt|Q-1b5|Q-7b|G-2b|G-9b|L-3b|L-8b|
> |:-:|:-:|:-:|:-:|:-:|:-:|:-:|:-:|
> |few shot|-|9.21|29.33|7.10|**54.23**|15.31|24.71|
> |CMC|Q-1b5|8.99|21.09|15.90|44.78|14.96|19.99|
> ||G-2b|1.03|7.19|29.78|39.96|1.71|1.93|
> ||L-3b|13.79|22.13|18.42|42.18|18.51|23.51|
> |TOA|Q-1b5|**53.81**|**30.99**|21.08|9.58|29.24|26.94|
> ||G-2b|23.12|12.47|**54.53**|24.31|22.64|15.80|
> ||L-3b|27.48|21.79|24.79|14.05|**54.51**|**45.24**|
>
>
> **Analysis of the zero-shot and few-shot performance:**
> 1. Using NER as an example, LMs may not fully grasp certain nuances of the task. For instance (Cases 1 & 2), they often incorrectly predict pronouns like 'I' or 'you' as named entities. They also exhibit counterintuitive errors, such as labeling phrases like 'our perspective' as an 'EVENT'.
> 2. LLMs sometimes generate outputs that seem reasonable but diverge from dataset labels (Cases 3 & 4). This discrepancy resembles a mismatch with human labeling preferences. For example, where the dataset annotates 'Zhou' as 'Person', LMs prefer the full span 'Professor Zhou'. Similarly, LMs may classify 'mobile phone SMS' as 'PRODUCT', while the original labels do not. These represent comprehensible deviations from dataset conventions, rather than fundamental errors.
> 3. Few-shot prompting improves modestly over zero-shot, but gains remain limited. TOA typically outperforms both (Table 1). Smaller models (Qwen-1.5B, Gemma-2B, LLaMA-3B) benefit more from few-shot examples than larger counterparts (Qwen-7B, Gemma-9B, LLaMA-8B), likely because the limited information from only five examples adds negligible value to large models with extensive pre-trained knowledge.
>
> **case study:**
>
> For Case 1, we show the full prompt (including the template from Section K). For other cases, we display only the input sentence and model output; their prompts were generated by inserting the input after the template’s 'Sentence:'.
>
> Case1（Q-7b）:
> > **Prompt**:
> > Act as a named entity recognition tagging tool. Find all entities and their classes in a sentence according to these rules:
> > 1.Choose the correct named entity class from \[PERSON, NORP, FACILITY, ORGANIZATION, Geo-Political Entity, LOCATION, PRODUCT, DATE, TIME, PERCENT, MONEY, QUANTITY, ORDINAL, CARDINAL, EVENT, WORK_OF_ART, LAW, LANGUAGE\].
> > 2.Do not explain or add extra text.
> > Sentence: Well , I especially want to know , ha , how the two of you found out the news on the day of the accident ?
> > Response as tuples, and each tuple must have exactly two elements: first element is the named entity text (as a string), second element is the named entity class (as a string), e.g. (\<entity1\>, \<class1\>), (\<entity2\>, \<class2\>), ...
> > Response:
> >
> > **Sentence**: Well , I especially want to know , ha , how the two of you found out the news on the day of the accident ?
> >
> > **Lables**: (two, CARDINAL)
> >
> > **Outputs**: (I, PERSON), (two of you, GROUP), (the news, PRODUCT), (the day, DATE), (the accident, EVENT)
>
> Case2（L-8b）:
> > **Sentence**: From our perspective as media , I think it was precisely by going through those breaking - in , trials that we could be so calm in handling such an emergency today , ah , and give everyone the most effective announcements and notifications , and instructions .
> >
> > **Lables**: (today, DATE)
> >
> > **Outputs**: ('our perspective', 'EVENT') ('as media', 'FACILITY') ('I', 'PERSON') ('it', 'PERSON') ('those breaking - in , trials', 'EVENT') ('today', 'DATE') ('ah', 'PERSON')
>
> Case3（Q-7b）:
> > **Sentence**: Well , like Professor Zhou , I also received this news , ha , through a mobile phone SMS .
> >
> > **Lables**: (Zhou, PERSON)
> >
> > **Outputs**: ('Professor Zhou', PERSON), ('news', EVENT), ('mobile phone SMS', PRODUCT)
>
> Case4（L-8b）:
> > **Sentence**: I think , in comparison to China , we should say that urbanization in foreign countries developed earlier and is more widespread .
> >
> > **Lables**: (China, Geo-Political Entity)
> >
> > **Outputs**: ('China', 'Geo-Political Entity'),('urbanization', 'EVENT'),('foreign countries', 'LOCATION')
>
>
> **Analysis of CMC Results:**
>
> 1. CMC enhances an LM's performance by training a delta LM that concurrently reasons on the same input text alongside the source LM and integrates their output logits. Furthermore, this delta LM can be applied to enhance other target LMs, demonstrating promising results in instruction-following tasks. However, they utilized a delta LM (TinyLlama) and source LM (LLaMA2-7B) with identical tokenizers, and they only explored scenarios where the delta LM's tokenizer differed from the target LM's (e.g., LLaMA2-70B, Mistral-7B) via token mapping based on metrics like edit distance. We observed that when the source LM (e.g., Qwen-1.5B) and its delta LM (e.g., TinyLlama) employ different tokenizers, the knowledge transfer from the delta LM to target LMs faces challenges, leading to suboptimal performance. This performance gap likely arises from mismatches in tokenizer mapping between the source/target LMs and the delta LM, which can cause a distributional discrepancy in the tokens processed by the delta LM during both training and inference. Since the OLA of two LMs with different tokenizers remain similar, we did not encounter this problem.
> 2. We found that incorporating CMC may introduce previously absent formatting issues, such as nested parentheses, extra spaces, or superfluous commas. These artifacts somewhat lowered its overall score.
>
> # 3. TOA require training independent adapters for downstream tasks. (w3)
>
> Yes, but learning separate adapters for different downstream tasks is a common practice in adapter-based methods [3,4]. Moreover, **once TOA is trained, it can be directly used by other LMs without retraining, making it efficient.**
>
> # 4. Performance Degradation After Transferring TOA. (w4)
>
> Training-free cross-model transfer is profoundly challenging due to the absence of target model information. This inevitably leads to performance degradation after TOA transfer; nevertheless, **its performance remains superior to the target model's zero-shot capability, indicating successful knowledge transfer to some extent.** Notably, this limitation also persists in other transfer methods [2,5].
>
> # 5. Unclear Expression (w5)
>
> **Thank you for the feedback. We will clarify this expression in subsequent revisions.**
>
> By 'outliers,' we refer to points within a row that are significantly larger than the others (Line 583). In the example in Sec. 3.2, only the maximum value in each row is considered an outlier. For ease of understanding, we explicitly stated to 'set the maximum value to zero'.
> In Sec. B, multiple outliers may exist per row. Therefore, we used the term 'outliers'.
>
> We will remove the preposition 'of' from the phrase 'The diagonal features of…'.
>
> # 6. Metrics for Measuring OLA Similarity (q3)
>
> **We used a 2D similarity metric (visual model, SSIM) instead of 1D vector metrics because different LMs employ distinct tokenizers.** Even when resized to identical dimensions, corresponding pixels do not consistently represent the same key-value pairs (Figure 3). Thus, a 2D structural indicator is more appropriate than pixel-to-pixel metrics.
>
> **For comprehensive evaluation, we employed simpler vector similarity metrics in our image similarity retrieval experiment (Section 3.2.2), including the Cosine Similarity and Mean Squared Error (MSE).** It can be seen from the results in Tables 2 and 3 below that all tested metrics reveal the cross-model similarity of OLA.
>
> A more detailed discussion can be found in R4-#2.
>
> **Table 2: First-order OLA cross-model similarity using Cosine Similarity (Experiment settings match main text Table 2)**
>
> |Src\Tgt|Q-1b5|G-2b|L-3b|
> |:-:|:-:|:-:|:-:|
> |Q-1b5|-|83.40/88.70|94.20/96.80|
> |G-2b|78.70/85.80|-|97.40/98.60|
> |L-3b|90.60/93.90|96.70/97.80|-|
>
> **Table 3: Results using the MSE metric.**
>
> |Src\Tgt|Q-1b5|G-2b|L-3b|
> |:-:|:-:|:-:|:-:|
> |Q-1b5|-|84.20/89.20|94.50/97.10|
> |G-2b|78.40/85.10|-|98.00/98.80|
> |L-3b|88.40/92.50|97.00/98.10|-|
>
> # References:
>
> [1] Relative representations enable zero-shot latent space communication.
>
> [2] Cross-model Control: Improving Multiple Large Language Models in One-time Training.
>
> [3] Scalable Language Model with Generalized Continual Learning.
>
> [4] Boosting Continual Learning of Vision-Language Models via Mixture-of-Experts Adapters.
>
> [5] Zero-Shot Continuous Prompt Transfer: Generalizing Task Semantics Across Language Models.

---

> ### Author Response · Authors · 2025-08-07
>
> Dear Reviewer aRHQ,
>
> Thank you for your detailed review and valuable suggestions. As the discussion period is ending soon, we warmly invite you to take a final look at our rebuttal.
>
> To briefly summarize our response:
>
> 1. ​We discussed the limitations of applying TOA to semantics-dependent tasks and potential improvement methods.​​
> 2. We added few-shot prompting and Cross-Model Control as comparison methods; TOA still achieved the best performance. Furthermore, we analyzed the reasons for the LLM's poor performance through case studies.​​
> 3. ​We discussed that TOA requiring training an independent adapter for each task is a common practice. Moreover, due to its cross-model transferability, TOA only needs to be trained once and can be used directly by other models, demonstrating its efficiency.​
> 4. ​We discussed that as training-free adapter transfer is a very challenging task, some performance degradation after TOA transfer is difficult to avoid. However, TOA's performance being better than zero-shot demonstrates that TOA effectively achieves knowledge transfer.​​
> 5. ​We thank you for pointing out the unclear expressions in our writing; we will revise them accordingly.​​
> 6. ​We explained the rationale for using the 2D similarity metric (visual model, SSIM) and supplemented it with simpler vector similarity metrics. Experiments further confirm the existence of cross-model similarity in OLA.​​
>
> Thank you again for your thoughtful feedback!
>
> Best regards,
>
> Paper #11941 Authors

---

### Note · Authors · 2025-08-12

Dear AC and Reviewers,

Thank you sincerely for the time and effort you have invested in the review process. In our final remarks, we wish to reiterate the contributions and limitations of our work to facilitate your assessment.

**Contributions:​​**

1. We propose Order-Level Attention (OLA), which unifies the attention of different LMs into comparable representations.
2. We propose two key findings:
   1. The OLA of the same order reveals significant similarity across distinct LMs given identical text inputs.​
   2. OLA inherently captures syntactic knowledge.​
3. Building on these findings, we introduce Transferable OLA Adapter (TOA), enabling training-free cross-LM adapter transfer.

We suggest that this work provides an innovative perspective for studying LMs by revealing commonalities in their attention patterns.This could support future research on LM interpretability and knowledge transfer. Furthermore, the TOA demonstrates a potential application of OLA and represents a step forward in enabling adapter transfer across LMs.

**​Limitations & Future Work:​​**

Since OLA is derived from attention maps, it suffers from information loss compared to features from traditional learnable encoders. OLA primarily captures syntactic structure and contains limited semantic information, restricting TOA's performance on semantics-dependent tasks. We are investigating the integration of latent space communication techniques [1] into OLA by representing each word's semantics through its relevance to anchor words. This approach holds promise to enhance OLA's semantic capacity while preserving its cross-LM consistency.

**​Gratitude:​​**

Finally, we sincerely thank the reviewers for their positive feedback and constructive suggestions. We are particularly encouraged by praise of the **"novel perspective" and "strong breadth and depth" (LYHK)**, remarks that **"thorough and well designed experiments" and "simple yet effective methods" (uyDY)**, assessment that **"this is a good paper and appears ready for publication" (dT4c)**, and recognition of **"far-reaching research questions" and "comprehensive and insightful research methods" (U4s6)**. We are also grateful for valuable suggestions such as strengthening limitations discussion, clarifying the OLA-syntax link, and expanding related work. These will significantly enhance the paper.

Best regards,

Paper #11941 Authors

[1] Relative representations enable zero-shot latent space communication.

---

### Decision · Program_Chairs · 2025-09-17

**Decision:**

Accept (poster)

**Comment:**

The paper investigates commonalities in the representations between different LMs. It proposes Order-Level Attention (OLA), which allows authors to compare attention modules from different LLMs. It is argued that OLAs are similar across different LMs. Based on these findings a training free cross-LMs adaptor (TOA) is proposed. These techniques are studied empirically.

Generally, reviewers believe that the paper is ready to be accepted. In the final version the authors should improve the related work section and better contextualize their contribution in the broader research community, per Reviewer dT4c request.